# Quantum Work Capacitances: ultimate limits for energy extraction on noisy quantum batteries

Salvatore Tirone,[1, *] Raffaele Salvia,[2, †] Stefano Chessa,[2, 3, ‡] and Vittorio Giovannetti[2]

[1]*Scuola Normale Superiore, I-56126 Pisa, Italy*
[2]*NEST, Scuola Normale Superiore and Istituto Nanoscienze-CNR, I-56126 Pisa, Italy*
[3]*Electrical and Computer Engineering, University of Illinois Urbana-Champaign, Urbana, Illinois, 61801, USA*

We present a theoretical analysis of the energy recovery efficiency for quantum batteries composed of many identical quantum cells undergoing noise. While the possibility of using quantum effects to speed up the charging processes of batteries have been vastly investigated, In order to traslate these ideas into working devices it is crucial to assess the stability of the storage phase in the quantum battery elements when they are in contact with environmental noise. In this work we formalize this problem introducing a series of operationally well defined figures of merit (the work capacitances and the Maximal Asymptotic Work/Energy Ratios) which gauge the highest efficiency one can attain in recovering useful energy from quantum battery models that are formed by large collections of identical and independent elements (quantum cells or q-cells). Explicit evaluations of such quantities are presented for the case where the energy storing system undergoes through dephasing and depolarizing noise.

## I. INTRODUCTION

Quantum batteries are an emerging concept that aims to exploit quantum mechanical effects to improve the energy storage capabilities at the nanoscale. These devices employ quantum resources like coherence, entanglement and squeezing to boost properties like charging power, storage capacity and stability. Realizing practical quantum batteries will require the optimization of these quantum enhancements while accounting for the inevitable noise processes that will deteriorate their overall performance. The possibility of improving the energetic efficiency of present and future quantum devices [1–4] provides motivations for investigating the concepts of work and heat in the quantum realm [5] and for developing alternative, quantum-based architectures for energy storage [6, 7]. Such objects allocate energy in the states of externally controllable quantum systems, hereafter identified with the conventional name of Quantum Batteries, suggesting the possibility of achieving fast charging performances [7–12] by exploiting inherent quantum effects such as dynamical entanglement. However, as also happens in classical models, storing and recovering energy from a system are not necessarily equivalent processes: different states of a quantum battery which are associated with the same mean value of the stored energy may yield different throughput values in terms of the energy (or at least of the *useful* part of the energy, i.e. work) one can recover from them. On top of this, one has also to consider that in any realistic implementation quantum batteries will be inevitably subjected to environmental noise. This, as the first experimental demonstrations show [13–18], will typically mess up with the energy re-

covering process, not to mention the storage itself. To solve this last problem various stabilization schemes have been proposed, both passive and active [5, 19–27] and other instances of specific noise effects in quantum batteries have been studied, see e.g. [28–35]. Our work is complementary to these approaches and aims to provide a recipe to address arbitrary noise models: we propose to mitigate noise effects by initializing our quantum battery with the more error-resistant state possible. For this purpose we define various measures that quantify how a selected state of a quantum battery is resilient to the detrimental action of a given noise model and use them to identify the optimal candidates as input states for the energy recovery task. A similar endeavour was tackled in Ref. [36, 37] and mimics the optimization problem one faces in quantum communication [38–43] when designing information encoding strategies that are capable of mitigating the detrimental effects of dissipation and decoherence.

In contrast to existing measures focused on transient power enhancement, our approach emphasizes the long-term charging capabilities and energetic efficiency of noisy quantum batteries. Such new measures can incorporate constraints on resources like entanglement and non-local operations [44] and shed light on the usefulness of quantum resources such as coherence [45], providing guidance on optimal control. Furthermore, the framework we propose complements the existing literature on quantum advantages by providing means to directly compare the impact of different noise models on their realistic achievability.

Our analysis relies on functionals [36, 37], e.g. the ergotropy, that we leverage in order to quantify the maximum amount of useful energy (work) one can extract from a given state of a quantum system, depending on the resources that we're allowed to exploit for the task [46]. Among others, the *ergotropy* functional [47] is arguably the most widely studied. It represents the maximum mean value of extractable energy that one can get from

* salvatore.tirone@sns.it
† raffaele.salvia@sns.it
‡ schessa@illinois.edu

closed models where the only allowed operations are modulations of the system Hamiltonian (no interactions with external elements being allowed). We introduce figures of merit called "*quantum work capacitances*" and "*maximal asymptotic work/energy ratios*" (MAWER) to quantify the maximum extractable work per quantum cell for a given noise model under different resource contraints. For quantum batteries with a large number of cells, they quantify how the maximum achievable extractable work per cell approaches a limit called work capacitance. Intuitively, the capacitances gauge the batteries stability against noise during the charging, storage and discharging processes. More explicitly, the work capacitance gives the maximum stable energy density that can be stored per cell, while the MAWER quantifies the battery's energy efficiency as the number of cells increases. So together they characterize the scalable work output and efficiency.

To show the capabilties of this framework, we provide the first attempt of evaluation of the work capacitance and MAWER for two noise models: dephasing and depolarizing noise. For dephasing, we show that entanglement provides no advantage, while for depolarizing noise global operations are beneficial but entanglement is not. Our results help to identify optimal charging strategies to maximize extractable work. For instance, for dephasing noise, separable states perform just as well as entangled states; but for depolarizing noise, global operations give an advantage.

The the manuscript is organized as follows.

In Sec. II we set up the notation and introduce the key concepts of work extraction functionals. We formulate the constrained optimization problems that quantify the useful extractable work of a quantum battery under noise.

In Sec. III we specialize this theoretical framework to model quantum batteries made of many identical quantum cells undergoing local noise. The figures of merit called work capacitances and MAWERs are introduced to characterize the batteries scalable work output and efficiency.

In Sec. IV we establish some mathematical properties of the work extraction functionals that will simplify the subsequent analysis. These results provide general insights into the behavior of quantum batteries regardless of the specific noise model.

In Sec. V we apply the tools developed in the previous section to the analysis of two practical noise models - dephasing and depolarizing. Exact solutions for the work capacitance and MAWER reveal the existence of a quantum advantage by using quantum resources, and might provide guidance for optimizing real quantum battery designs.

Conclusions are drawn in Sec. VI, while technical derivations are moved into the Appendices.

## II. ENERGY STORAGE VS ENERGY RECOVERING IN NOISY QUANTUM SYSTEMS

This section is devoted to the introduction of the fundamental theoretical tools that will be used to characterize the energy recovery efficiency for noisy quantum battery models.

### A. Work extraction functionals for quantum systems

Consider a quantum system $Q$ represented by a $d$-dimensional Hilbert space $\mathcal{H}$ and by a Hamiltonian $\hat{H}$ whose ground state energy is assumed to be zero without loss of generality. Being $\hat{\rho}$ a state of $Q$, the energy it can store on average is given by the expectation value

$$\mathfrak{E}(\hat{\rho}; \hat{H}) := \mathrm{Tr}\left[\hat{\rho}\hat{H}\right] . \qquad (1)$$

The energy that we can recover from $\hat{\rho}$ does not necessarily equal $\mathfrak{E}(\hat{\rho}; \hat{H})$ and strongly depends on the process that is implemented for the task. Following [36, 46, 47], we compute it in terms of functionals $\mathcal{W}(\hat{\rho}; \hat{H})$ describing the maximum amount of useful energy (work) that an agent can recover from $Q$ for a given set of allowed operations.

*a. Ergotropy:–* The first (and most fundamental) of these quantities is the ergotropy functional:

$$\mathcal{E}(\hat{\rho}; \hat{H}) := \max_{\hat{U} \in \mathbb{U}(d)} \left\{ \mathfrak{E}(\hat{\rho}; \hat{H}) - \mathfrak{E}(\hat{U}\hat{\rho}\hat{U}^\dagger; \hat{H}) \right\} . \qquad (2)$$

As mentioned in the introduction, the ergotropy represents the maximum energy that an agent can get from $\hat{\rho}$ when the allowed transformations are given by the set $\mathbb{U}(d)$ of the unitary operators acting on $Q$. The right-hand-side of Eq. (2) admits a closed expression in terms of the passive counterpart of $\hat{\rho}$. Explicitly, the density matrix $\hat{\rho}_{\mathrm{pass}}$ is obtained by operating on $\hat{\rho}$ via a unitary rotation that transforms its eigenvectors $\{|\lambda_\ell\rangle\}_\ell$ into the eigenvectors $\{|E_\ell\rangle\}_\ell$ of $\hat{H}$, matching the corresponding eigenvalues in reverse order, i.e.

$$\left. \begin{array}{l} \hat{\rho} = \sum_{\ell=1}^d \lambda_\ell |\lambda_\ell\rangle\langle\lambda_\ell| \\[2mm] \hat{H} = \sum_{\ell=1}^d E_\ell |E_\ell\rangle\langle E_\ell| \end{array} \right\} \mapsto \hat{\rho}_{\mathrm{pass}} := \sum_{\ell=1}^d \lambda_\ell |E_\ell\rangle\langle E_\ell| , \qquad (3)$$

where for all $\ell = 1, \cdots, d-1$, we set $\lambda_\ell \geq \lambda_{\ell+1}$ and $E_\ell \leq E_{\ell+1}$. We can then write

$$\mathcal{E}(\hat{\rho}; \hat{H}) = \mathfrak{E}(\hat{\rho}; \hat{H}) - \mathfrak{E}(\hat{\rho}_{\mathrm{pass}}; \hat{H}) = \mathfrak{E}(\hat{\rho}; \hat{H}) - \sum_{\ell=1}^d \lambda_\ell E_\ell . \qquad (4)$$

*b. Total-ergotropy:–* The second work extraction functional $\mathcal{W}(\hat{\rho}; \hat{H})$ we aim to study is the total-ergotropy $\mathcal{E}_{\mathrm{tot}}(\hat{\rho}; \hat{H})$. The total ergotropy is a regularized version of the ergotropy $\mathcal{E}(\hat{\rho}; \hat{H})$ that emerges when considering scenarios where one has at disposal an arbitrary large number of identical copies of the input state $\hat{\rho}$. Formally it is defined as

$$\mathcal{E}_{\mathrm{tot}}(\hat{\rho}; \hat{H}) := \lim_{n \to \infty} \frac{\mathcal{E}(\hat{\rho}^{\otimes n}; \hat{H}^{(n)})}{n} , \qquad (5)$$

where for a fixed number of copies $n$, $\hat{H}^{(n)}$ is the total Hamiltonian of the $n$ copies of the system obtained by assigning to each of them the same $\hat{H}$ (no interactions being included). One can show [6, 48] that the limit in Eq. (5) exists and corresponds to the maximum of $\frac{\mathcal{E}(\hat{\rho}^{\otimes n}; \hat{H}^{(n)})}{n}$ with respect to all possible $n$, implying in particular that $\mathcal{E}_{\mathrm{tot}}(\hat{\rho}; \hat{H})$ is at least as large as $\mathcal{E}(\hat{\rho}; \hat{H})$ (in systems of dimension $d = 2$ we have $\mathcal{E}(\hat{\rho}; \hat{H}) = \mathcal{E}_{\mathrm{tot}}(\hat{\rho}; \hat{H})$ for all inputs [46], while in general the strict inequality holds for dimension greater than 2). The total ergotropy $\mathcal{E}_{\mathrm{tot}}(\hat{\rho}; \hat{H})$, as shown in [6], can be expressed via a single letter formula that mimics Eq. (4):

$$\mathcal{E}_{\mathrm{tot}}(\hat{\rho}; \hat{H}) = \mathfrak{E}(\hat{\rho}; \hat{H}) - \mathfrak{E}_{\mathrm{GIBBS}}^{(\beta_\star)}(\hat{H}) , \qquad (6)$$

where for $\beta \geq 0$ and $Z_\beta(\hat{H}) := \mathrm{Tr}[e^{-\beta \hat{H}}] = \sum_{\ell=1}^{d} e^{-\beta E_\ell}$ we can express

$$\mathfrak{E}_{\mathrm{GIBBS}}^{(\beta)}(\hat{H}) := \mathfrak{E}(\hat{\rho}_\beta; \hat{H}) = -\frac{d}{d\beta} \ln Z_\beta(\hat{H}) , \qquad (7)$$

and the mean energy of the thermal Gibbs state

$$\hat{\rho}_\beta := e^{-\beta \hat{H}} / Z_\beta(\hat{H}) , \qquad (8)$$

of $Q$ with effective inverse temperature $\beta$. Finally, $\beta_\star$ is chosen so that $\hat{\rho}_{\beta_\star}$ has the same von Neumann entropy of $\hat{\rho}$, i.e. $S_{\beta_\star} = S(\hat{\rho}) := -\mathrm{Tr}[\hat{\rho} \ln \hat{\rho}]$, with

$$S_\beta := -\mathrm{Tr}[\hat{\rho}_\beta \ln \hat{\rho}_\beta] = -\beta \frac{d}{d\beta} \ln Z_\beta(\hat{H}) + \ln Z_\beta(\hat{H}) . \quad (9)$$

The state $\hat{\rho}_\beta$ has by construction the same entropy as $\hat{\rho}$, and it is the state with the minimum energy among the set of states with entropy $S(\rho)$. So the total ergotropy represents the maximum energy that can be extracted from the system by any transformation that keeps the entropy constant. The total ergotropy $\mathcal{E}_{\mathrm{tot}}$ it is monotonically decreasing with the increasing system entropy for states which have the same mean energy, i.e. given two iso-energetic states $\hat{\rho}_1$ and $\hat{\rho}_2$ such that $S(\hat{\rho}_1) \leq S(\hat{\rho}_2)$ it always holds that

$$\mathcal{E}_{\mathrm{tot}}(\hat{\rho}_1; \hat{H}) \geq \mathcal{E}_{\mathrm{tot}}(\hat{\rho}_2; \hat{H}) . \qquad (10)$$

*c. Non-equilibrium free Energy:–* The non-equilibrium free energy defines the maximum amount of work obtainable from $\hat{\rho}$ if $Q$ can be put in thermal contact with an external bath of fixed inverse temperature $\beta \geq 0$. The non-equilibrium free energy of a state is defined as

$$\mathcal{F}_\beta(\hat{\rho}; \hat{H}) := \mathfrak{E}(\hat{\rho}; \hat{H}) - \frac{1}{\beta} S(\hat{\rho}) , \qquad (11)$$

and for thermal equilibrium states in the form (8) it takes the form $\mathcal{F}_\beta(\hat{\rho}_\beta; \hat{H}) = -\frac{1}{\beta} \log Z_\beta(\hat{H})$. The resulting work extraction functional $\mathcal{W}(\hat{\rho}; \hat{H})$ writes then as the free energy difference between the initial state $\hat{\rho}$ and the thermal state with inverse temperature $\beta$. That is

$$\begin{aligned} \mathcal{W}_\beta(\hat{\rho}) &:= \mathcal{F}_\beta(\hat{\rho}; \hat{H}) - \mathcal{F}_\beta(\hat{\rho}_\beta; \hat{H}) \\ &= \mathcal{F}_\beta(\hat{\rho}; \hat{H}) + \frac{\log Z_\beta(\hat{H})}{\beta} . \end{aligned} \qquad (12)$$

*d. Local Ergotropy:–* Our final work extraction functional is introduced for the cases where, as in the quantum battery models we'll consider in Sec. III, $Q$ is a composite system formed by a collection of individual elements. In this context we define the local ergotropy $\mathcal{E}_{\mathrm{loc}}(\hat{\rho}; \hat{H})$ as the maximum work obtainable from $\hat{\rho}$ when the unitary operations that can be performed are bound to be local with respect to the many-body partitions of the system. Therefore it can be computed by solving the following maximization problem

$$\mathcal{E}_{\mathrm{loc}}(\hat{\rho}; \hat{H}) := \max_{\hat{U} \in \mathbb{U}_{\mathrm{loc}}(d)} \left\{ \mathfrak{E}(\hat{\rho}; \hat{H}) - \mathfrak{E}(\hat{U} \hat{\rho} \hat{U}^\dagger; \hat{H}) \right\} , \quad (13)$$

with $\mathbb{U}_{\mathrm{loc}}(d)$ being the subset of $\mathbb{U}(d)$ formed by local transformations. Apart from some special cases [49] no closed expressions are known for Eq. (13): by construction it is clear however that $\mathcal{E}_{\mathrm{loc}}(\hat{\rho}; \hat{H})$ is certainly not larger than $\mathcal{E}(\hat{\rho}; \hat{H})$.

## B. Optimal output work extraction functionals

By construction all the quantities $\mathcal{W}(\hat{\rho}; \hat{H})$ defined in the previous sections are positive semidefinite and are strictly smaller than $\mathfrak{E}(\hat{\rho}; \hat{H})$, meaning that not all the energy stored in $\hat{\rho}$ can be recovered in general (for $\mathcal{E}(\hat{\rho}; \hat{H})$ and $\mathcal{E}_{\mathrm{tot}}(\hat{\rho}; \hat{H})$ this happens only if $\hat{\rho}$ is a pure state [47]). It is also worth observing that $\mathcal{E}(\hat{\rho}; \hat{H})$, $\mathcal{E}_{\mathrm{tot}}(\hat{\rho}; \hat{H})$, and $\mathcal{F}_\beta(\hat{\rho}; \hat{H})$ are invariant under the group of energy preserving unitary transformations

$$\mathbb{U}_{EP} := \{ \hat{V} \in \mathbb{U}(d) : [\hat{V}, \hat{H}] = 0 \} , \qquad (14)$$

i.e. the set formed by operators that in the eigen-energy basis of the model (see Eq. (3)) can be expressed as

$$\hat{V} = \sum_{\ell=1}^{d} e^{i\phi_\ell} |E_\ell\rangle\langle E_\ell| , \qquad (15)$$

with $\phi_\ell$ real parameters. Indeed since $\mathfrak{E}(\hat{\rho}_{\mathrm{pass}}; \hat{H})$ and $\mathfrak{E}(\hat{\rho}_\beta; \hat{H})$ are invariant under *any* unitary transformation

acting on $\hat{\rho}$, and $\mathfrak{E}(\hat{\rho}; \hat{H})$ is by construction invariant under all elements of $\mathbb{U}_{EP}$, for the ergotropy, the total-ergotropy, and the non-equilibrium free-energy we can write

$$\mathcal{W}(\hat{\rho}; \hat{H}) = \mathcal{W}(\hat{V}\hat{\rho}\hat{V}^\dagger; \hat{H}) , \qquad \forall \hat{V} \in \mathbb{U}_{EP} . \quad (16)$$

Equation (16) implies in particular that these work functionals behave as constants of motion under the evolution induced by the system Hamiltonian $\hat{H}$. The same property holds true also for the local ergotropy, at least for the special cases where the Hamiltonian of the joint system does not include interactions (this follows from the fact that Eq. (16) applies to $\mathcal{E}_{\text{loc}}(\hat{\rho}; \hat{H})$ if we restrict $\hat{V}$ to the subset of local elements of $\mathbb{U}_{EP}$). The situation of course changes when the dynamics of the system is perturbed by some external disturbance, e.g. induced by a coupling of $Q$ with an external bath. Under these conditions there is no guarantee that the extractable energy one could get at the beginning of the process would be the same as the one obtainable from a deteriorated version $\hat{\rho}_{\text{out}}$ of the initial state $\hat{\rho}$, with $\hat{\rho}_{\text{out}} := \Lambda(\hat{\rho})$ and being $\Lambda$ the completely positive and trace-preserving (CPTP) channel [38] describing the action of the noisy evolution. To evaluate the efficiency of the energy release process in such a context we compute the maximum values that our work extraction functionals assume after the action of the noisy channel $\Lambda$, under the condition that the input states $\hat{\rho}$ are constrained to some fixed subset of allowed configurations. In particular, given $E \in [0, E_d]$, being $E_d$ the highest eigenvalue of the considered Hamiltonian, we introduce the energy shell subsets

$$\overline{\mathfrak{S}}_E := \left\{ \hat{\rho} : \mathfrak{E}(\hat{\rho}; \hat{H}) = E \right\} , \quad (17)$$

$$\mathfrak{S}_E := \left\{ \hat{\rho} : \mathfrak{E}(\hat{\rho}; \hat{H}) \leq E \right\} = \bigcup_{0 \leq E' \leq E} \overline{\mathfrak{S}}_{E'} , \quad (18)$$

and define the quantities

$$\overline{\mathcal{W}}(\Lambda; E) := \max_{\hat{\rho} \in \overline{\mathfrak{S}}_E} \mathcal{W}(\Lambda(\hat{\rho}); \hat{H}) , \quad (19)$$

$$\mathcal{W}(\Lambda; E) := \max_{\hat{\rho} \in \mathfrak{S}_E} \mathcal{W}(\Lambda(\hat{\rho}); \hat{H}) \quad (20)$$
$$= \max_{0 \leq E' \leq E} \overline{\mathcal{W}}(\Lambda; E') \geq \overline{\mathcal{W}}(\Lambda; E) ,$$

which gauge the maximum output work of the model per fixed input energy. Specifically $\overline{\mathcal{W}}(\Lambda; E)$ assumes a sharp energy constraint that allows only states with energy that exactly matches the threshold $E$, while $\mathcal{W}(\Lambda; E)$ allows for the possibility of using also initial configurations with input energy smaller than $E$. For the special case where $Q$ is a composite system we will also consider the possibility of further restricting the allowed states to non-entangled configurations, defining

$$\overline{\mathcal{W}}_{\text{sep}}(\Lambda; E) := \max_{\hat{\rho}_{\text{sep}} \in \overline{\mathfrak{S}}_E} \mathcal{W}(\Lambda(\hat{\rho}_{\text{sep}}); \hat{H}) , \quad (21)$$

$$\mathcal{W}_{\text{sep}}(\Lambda; E) := \max_{\hat{\rho}_{\text{sep}} \in \mathfrak{S}_E} \mathcal{W}(\Lambda(\hat{\rho}_{\text{sep}}); \hat{H}) \quad (22)$$
$$= \max_{0 \leq E' \leq E} \overline{\mathcal{W}}_{\text{sep}}(\Lambda; E') ,$$

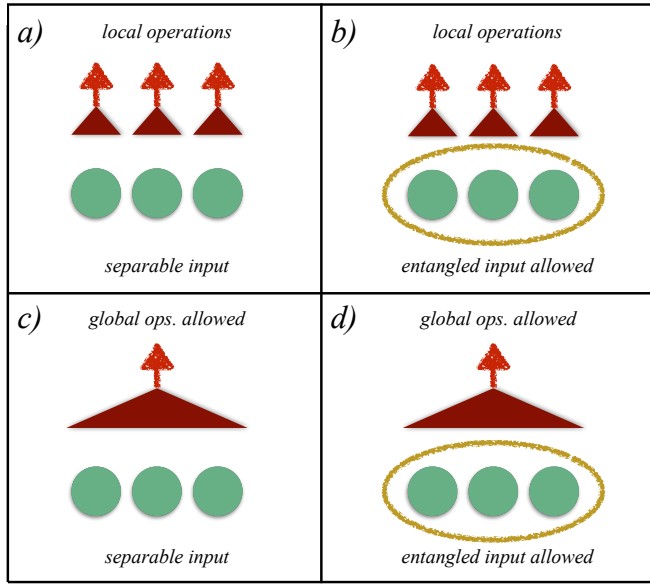

FIG. 1. Schematic representation of different work extraction protocols for multipartite models. In the scheme the green elements represent the different subsystems that compose the quantum battery, while red elements describe the operations being performed to recover the stored energy after the noise has perturbed the system. Panel a): the energy is charged on separable states of the q-cells (no entanglement), and it is recovered using only local operations. Panel b) entangled input states are allowed, but the energy release process is still mediated by local operations. Panel c) separable input states are used as input (no entanglement) but global operations are allowed. Panel d) completely unconstrained scenario: entanglement is allowed at the state preparation stage and global operations are permitted. In terms of the of the classification introduced in Sec. III A, $C_{\text{sep,loc}}(\Lambda; \mathfrak{e})$ of Eq. (46) is the capacitance of the scheme a); $C_{\text{loc}}(\Lambda; \mathfrak{e})$ of Eq. (45) is the one of the scheme b); $C_{\text{sep}}(\Lambda; \mathfrak{e})$ of Eq. (44) is the one for c); and finally $C_{\mathcal{E}}(\Lambda; \mathfrak{e})$ of Eq. (43) is the one for d).

with $\hat{\rho}_{\text{sep}}$ being the separable states contained in $\overline{\mathfrak{S}}_E$ and $\mathfrak{S}_E$. As graphically shown in Fig. 1 this allows us to identify four different scenarios which, similarly to what happens in quantum metrology and quantum communication [50, 51], are separated in terms of the locality constraints that are assumed at the level of state preparation and at the level of the output states manipulation processes.

## III. ENERGY RECOVERY FROM ARRAYS OF IDENTICAL (NOISY) Q-CELLS

We now focus on the case of quantum battery models consisting of a collection of $n$ identical and independent elements (q-cells). Each cell us represented by a $d$-dimensional Hilbert space $\mathcal{H}$ and characterized by a local Hamiltonian $\hat{h} := \sum_{i=1}^d \epsilon_i |i\rangle\langle i|$ with ordered eigenvalues

$\epsilon_1 \leq \epsilon_2 \leq ... \leq \epsilon_d$ that, via proper reshifting and renormalization, can be assumed to fulfil the conditions $\epsilon_1 = 0$ and $\epsilon_d = 1$. Similarly to what was done for the single q-cell scenario discussed before we can now address the multi-cell protocols. Here $\hat{\rho}^{(n)}$ will be a joint (possibly correlated) state of the quantum battery with $\hat{\rho}_j^{(n)}$ the reduced density matrix of the $j$-th q-cell, $\hat{H}^{(n)} := \sum_{j=1}^n \hat{h}_j$ will be instead the quantum battery Hamiltonian with $\hat{h}_j$ local Hamiltonian terms (notice that with the convention we have fixed $\mathfrak{E}(\hat{\rho}^{(n)}; \hat{H}^{(n)}) \in [0, n]$). We can consequently evaluate the energy that the battery stores on average as

$$\mathfrak{E}(\hat{\rho}^{(n)}; \hat{H}^{(n)}) := \mathrm{Tr}\left[\hat{\rho}^{(n)} \hat{H}^{(n)}\right] = \sum_{j=1}^n \mathrm{Tr}\left[\hat{\rho}_j^{(n)} \hat{h}_j\right], \quad (23)$$

As a noise model for the whole system we'll assume each q-cell undergoing the same CPTP transformation. Given $\hat{\rho}^{(n)}$ the input state of the quantum battery, its output will be described by the density matrix

$$\hat{\rho}_{\mathrm{out}}^{(n)} = \Lambda^{\otimes n}(\hat{\rho}^{(n)}). \quad (24)$$

With such a choice our energy-constrained figures of merit in Eq. (19) and Eq. (21) rewrite as

$$\mathcal{W}^{(n)}(\Lambda; E) := \max_{\hat{\rho}^{(n)} \in \mathfrak{S}_E^{(n)}} \mathcal{W}(\Lambda^{\otimes n}(\hat{\rho}^{(n)}); \hat{H}^{(n)}), \quad (25)$$

$$\mathcal{W}_{\mathrm{sep}}^{(n)}(\Lambda; E) := \max_{\hat{\rho}_{\mathrm{sep}}^{(n)} \in \mathfrak{S}_E^{(n)}} \mathcal{W}(\Lambda^{\otimes n}(\hat{\rho}_{\mathrm{sep}}^{(n)}); \hat{H}^{(n)}), \quad (26)$$

where as usual $\mathcal{W}$ stands for one of our four work extraction functionals (i.e. ergotropy, total-ergotropy, non-equilibrium free energy and local-ergotropy), and where given $E \in [0, n]$, we have $\mathfrak{S}_E^{(n)} := \left\{\hat{\rho}^{(n)} : \mathfrak{E}(\hat{\rho}^{(n)}; \hat{H}^{(n)}) \leq E\right\}$. In a similar fashion we also define $\overline{\mathcal{W}}^{(n)}(\Lambda; E)$, and $\overline{\mathcal{W}}_{\mathrm{sep}}^{(n)}(\Lambda; E)$ as the associated quantities obtained by restricting the optimization only over input that exactly match the energy constraint $E$, i.e. over the elements of the subset $\overline{\mathfrak{S}}_E^{(n)} := \left\{\hat{\rho}^{(n)} : \mathfrak{E}(\hat{\rho}^{(n)}; \hat{H}^{(n)}) = E\right\}$. In the following we specialize the definitions (25) and (26) to the specific figures of merit that we will use in the following sections:

$$\mathcal{E}^{(n)}(\Lambda; E) := \max_{\hat{\rho}^{(n)} \in \mathfrak{S}_E^{(n)}} \mathcal{E}(\Lambda^{\otimes n}(\hat{\rho}^{(n)}); \hat{H}^{(n)}), \quad (27)$$

$$\overline{\mathcal{E}}^{(n)}(\Lambda; E) := \max_{\hat{\rho}^{(n)} \in \overline{\mathfrak{S}}_E^{(n)}} \mathcal{E}(\Lambda^{\otimes n}(\hat{\rho}^{(n)}); \hat{H}^{(n)}), \quad (28)$$

$$\mathcal{E}_{\mathrm{sep}}^{(n)}(\Lambda; E) := \max_{\hat{\rho}_{\mathrm{sep}}^{(n)} \in \mathfrak{S}_E^{(n)}} \mathcal{E}(\Lambda^{\otimes n}(\hat{\rho}_{\mathrm{sep}}^{(n)}); \hat{H}^{(n)}), \quad (29)$$

$$\overline{\mathcal{E}}_{\mathrm{sep}}^{(n)}(\Lambda; E) := \max_{\hat{\rho}_{\mathrm{sep}}^{(n)} \in \overline{\mathfrak{S}}_E^{(n)}} \mathcal{E}(\Lambda^{\otimes n}(\hat{\rho}_{\mathrm{sep}}^{(n)}); \hat{H}^{(n)}), \quad (30)$$

$$\mathcal{E}_{\mathrm{loc}}^{(n)}(\Lambda; E) := \max_{\hat{\rho}^{(n)} \in \mathfrak{S}_E^{(n)}} \mathcal{E}_{\mathrm{loc}}(\Lambda^{\otimes n}(\hat{\rho}^{(n)}); \hat{H}^{(n)}), \quad (31)$$

$$\overline{\mathcal{E}}_{\mathrm{loc}}^{(n)}(\Lambda; E) := \max_{\hat{\rho}^{(n)} \in \overline{\mathfrak{S}}_E^{(n)}} \mathcal{E}_{\mathrm{loc}}(\Lambda^{\otimes n}(\hat{\rho}^{(n)}); \hat{H}^{(n)}). \quad (32)$$

Finally for the local and separable case we define

$$\mathcal{E}_{\mathrm{sep,loc}}^{(n)}(\Lambda; E) := \max_{\hat{\rho}_{\mathrm{sep}}^{(n)} \in \mathfrak{S}_E^{(n)}} \mathcal{E}_{\mathrm{loc}}(\Lambda^{\otimes n}(\hat{\rho}_{\mathrm{sep}}^{(n)}); \hat{H}^{(n)}), \quad (33)$$

$$\overline{\mathcal{E}}_{\mathrm{sep,loc}}^{(n)}(\Lambda; E) := \max_{\hat{\rho}_{\mathrm{sep}}^{(n)} \in \overline{\mathfrak{S}}_E^{(n)}} \mathcal{E}_{\mathrm{loc}}(\Lambda^{\otimes n}(\hat{\rho}_{\mathrm{sep}}^{(n)}); \hat{H}^{(n)}), \quad (34)$$

For the sake of clarity we also show the definitions for the output total ergotropy, starting by definition (5) we have

$$\mathcal{E}_{\mathrm{tot}}^{(n)}(\Lambda; E) := \max_{\hat{\rho}^{(n)} \in \mathfrak{S}_E^{(n)}} \mathcal{E}_{\mathrm{tot}}(\Lambda^{\otimes n}(\hat{\rho}^{(n)}); \hat{H}^{(n)})$$
$$= \max_{\hat{\rho}^{(n)} \in \mathfrak{S}_E^{(n)}} \lim_{k \to \infty} \frac{\mathcal{E}([\Lambda^{\otimes n}(\hat{\rho}^{(n)})]^{\otimes k}; \hat{H}^{(nk)})}{k}, \quad (35)$$

$$\overline{\mathcal{E}}_{\mathrm{tot}}^{(n)}(\Lambda; E) := \max_{\hat{\rho}^{(n)} \in \overline{\mathfrak{S}}_E^{(n)}} \mathcal{E}_{\mathrm{tot}}(\Lambda^{\otimes n}(\hat{\rho}^{(n)}); \hat{H}^{(n)})$$
$$= \max_{\hat{\rho}^{(n)} \in \overline{\mathfrak{S}}_E^{(n)}} \lim_{k \to \infty} \frac{\mathcal{E}([\Lambda^{\otimes n}(\hat{\rho}^{(n)})]^{\otimes k}; \hat{H}^{(nk)})}{k}. \quad (36)$$

It is straightforward to see that for every $n$, $E$ and $\Lambda$ we have that $\mathcal{E}_{\mathrm{tot}}^{(n)}(\Lambda; E) \geq \mathcal{E}^{(n)}(\Lambda; E)$, but in App. A 2 we prove that $\lim_{n\to\infty} \mathcal{E}^{(n)}(\Lambda; E)/n = \lim_{n\to\infty} \mathcal{E}_{\mathrm{tot}}^{(n)}(\Lambda; E)/n$ for any quantum channel $\Lambda$.

Moreover, we also explicitly define the quantities related to the work extraction in presence of a thermal bath:

$$\mathcal{F}_\beta^{(n)}(\Lambda; E) := \max_{\hat{\rho}^{(n)} \in \mathfrak{S}_E^{(n)}} \mathcal{F}_\beta(\Lambda^{\otimes n}(\hat{\rho}^{(n)}); \hat{H}^{(n)}), \quad (37)$$

$$\overline{\mathcal{F}}_\beta^{(n)}(\Lambda; E) := \max_{\hat{\rho}^{(n)} \in \overline{\mathfrak{S}}_E^{(n)}} \mathcal{F}_\beta(\Lambda^{\otimes n}(\hat{\rho}^{(n)}); \hat{H}^{(n)}). \quad (38)$$

We remark again here that the quantities with the bar are calculated on states which have precisely an average energy of $E$, while the others are computed on states with average energy less or equal than $E$. So in the second case the set on which we maximize our figure of merit is strictly bigger.

## A. Asymptotic limits

While $E$ and $n$ can in general be treated as independent terms, when studying quantum battery models with a large number of q-cells it makes sense to envision two different scenarios. Intuitively, we can imagine a quantum battery factory whose aim is to produce asymptotically large quantities of batteries: the main realistic constraints would be associated to *state preparation/processing* and *amount of energy available*.

In the first case we'd might be interested in understanding the maximum amount of extractable work per single cell (or per channel uses, from the noisy channel perspective) not focusing on the amount of energy that goes in

the cells: if the battery state preparation/management is more challenging than gathering the energy, knowing the scaling w.r.t. $n$ could be of more practical notice.

In the second case instead, we'd probably be more interested in knowing the amount of work we can extract per unit of energy we input in the cells: if for practical reasons we have that preparing/managing the cells is easy, we can assume to be able to distribute the input energy on an arbitrarily large number of cells and in that case the knowledge of the scaling w.r.t. the amount of input energy might be of higher interest.

*a. Work Capacitances:–* The first scenario refers to the cases where the threshold $E$ is comparable to $n$. To characterize these protocols we study the quantities in Eq. (25) and Eq. (26) fixing the ratio $E/n$ to a constant value $\mathfrak{e} \in [0,1]$. In particular, given the work extraction functional $\mathcal{W}$, we define the associated *work capacitance* as

$$C_{\mathcal{W}}(\Lambda; \mathfrak{e}) \ := \ \limsup_{n \to \infty} \frac{\mathcal{W}^{(n)}(\Lambda; E = n\mathfrak{e})}{n} \ , \quad (39)$$

which gauges the maximum work that one can extract per q-cell element when, on average, each one stores up to a fraction $\mathfrak{e}$ of the total input energy. It should be noticed that, due to the energy rescaling we established in our analysis, one has that $C_{\mathcal{W}}(\Lambda; \mathfrak{e})$ is guaranteed to be non-negative and not larger than 1, i.e.

$$0 \leq C_{\mathcal{W}}(\Lambda; \mathfrak{e}) \leq 1 \ , \quad (40)$$

with $C_{\mathcal{W}}(\Lambda; \mathfrak{e}) = 0$ implying the impossibility of extracting useful energy from the quantum battery, and $C_{\mathcal{W}}(\Lambda; \mathfrak{e}) = 1$ corresponding to the ideal work extraction performances. Notice also that, given $\mathfrak{e}' \leq \mathfrak{e}$, since the hierarchic order $\mathfrak{S}_{E=n\mathfrak{e}'}^{(n)} \subseteq \mathfrak{S}_{E=n\mathfrak{e}}^{(n)}$ holds for all $n$, we have that $C_{\mathcal{W}}(\Lambda; \mathfrak{e})$ is a monotonically non-decreasing function of $\mathfrak{e}$

$$C_{\mathcal{W}}(\Lambda; \mathfrak{e}') \leq C_{\mathcal{W}}(\Lambda; \mathfrak{e}) \ , \quad (41)$$

with the maximum value

$$C_{\mathcal{W}}(\Lambda) := C_{\mathcal{W}}(\Lambda; \mathfrak{e} = 1) \ , \quad (42)$$

corresponding to case where the optimization of the work functional is performed dropping the input energy constraint.

For all of the functionals that we take into consideration the evaluation of Eq. (39) is challenging. This is due to the possibility of super-additive effects for quantum batteries made of quantum cells which are initially entangled which each other. Using definition (37) for the non-equilibrium free-energy we obtain, by virtue of Eq. (12):

$$C_{\beta}(\Lambda; \mathfrak{e}) \ := \ \limsup_{n \to \infty} \frac{\mathcal{F}_{\beta}(\Lambda; n\mathfrak{e})}{n} + \frac{\log Z_{\beta}(\hat{h})}{\beta} \ .$$

By inspection we notice that the calculation of the above expression is closely related to the minimal output entropy problem since the only non-trivial term is given by the output free energy.

In what follows we will focus in particular on the ergotropy capacitance, by definition (27)

$$C_{\mathcal{E}}(\Lambda; \mathfrak{e}) := \limsup_{n \to \infty} \frac{\mathcal{E}^{(n)}(\Lambda; n\mathfrak{e})}{n} \ . \quad (43)$$

To estimate the role played in energy release process by the different type of available resources (e.g. entangled inputs and work-extraction global operations), we also define a collection of constrained versions of $C_{\mathcal{E}}(\Lambda; \mathfrak{e})$. In particular, following the schematic of Fig. 1, we introduce the *separable-input ergotropic capacitance* using definition (29),(31) and (33)

$$C_{\text{sep}}(\Lambda; \mathfrak{e}) := \limsup_{n \to \infty} \frac{\mathcal{E}_{\text{sep}}^{(n)}(\Lambda; n\mathfrak{e})}{n} \ , \quad (44)$$

the *local ergotropy capacitance*

$$C_{\text{loc}}(\Lambda; \mathfrak{e}) := \limsup_{n \to \infty} \frac{\mathcal{E}_{\text{loc}}^{(n)}(\Lambda; n\mathfrak{e})}{n} \ , \quad (45)$$

and the *separable-input, local ergotropy capacitance*

$$C_{\text{sep,loc}}(\Lambda; \mathfrak{e}) := \limsup_{n \to \infty} \frac{\mathcal{E}_{\text{sep,loc}}^{(n)}(\Lambda, n\mathfrak{e})}{n} \ . \quad (46)$$

Simple resource counting arguments impose a (partial) hierarchy between these terms, i.e.

$$C_{\mathcal{E}}(\Lambda; \mathfrak{e}) \geq C_{\text{loc}}(\Lambda; \mathfrak{e}), C_{\text{sep}}(\Lambda; \mathfrak{e}) \geq C_{\text{sep,loc}}(\Lambda; \mathfrak{e}) \ . \quad (47)$$

Notice however that no ordering has been selected between $C_{\text{loc}}(\Lambda; \mathfrak{e})$ and $C_{\text{sep}}(\Lambda; \mathfrak{e})$ since a priori there is no clear indication of whether for a given channel $\Lambda$ the use of global resources is more beneficial at the state preparation stage or at the end of the energy release process.

As shown in App. A 1, the $\limsup_{n \to \infty}$ in the above definitions correspond to simple $\lim_{n \to \infty}$ or, equivalently, to $\sup_{n \geq 1}$. Most importantly in App. A 2 we prove that, despite the fact that for all finite $n$, $\frac{\mathcal{E}_{\text{tot}}(\Lambda^{\otimes n}(\hat{\rho}^{(n)}); \hat{H}^{(n)})}{n}$ is always greater than or equal to $\frac{\mathcal{E}(\Lambda^{\otimes n}(\hat{\rho}^{(n)}); \hat{H}^{(n)})}{n}$, in the large $n$ limit the gap between these two rates goes to zero. Accordingly, while in principle one can introduce total-ergotropy equivalents $C_{\text{tot}}(\Lambda; \mathfrak{e})$ and $C_{\text{tot,sep}}(\Lambda; \mathfrak{e})$ of $C_{\mathcal{E}}(\Lambda; \mathfrak{e})$ and $C_{\text{sep}}(\Lambda; \mathfrak{e})$ respectively, by replacing $\mathcal{E}$ with $\mathcal{E}_{\text{tot}}$ in the right-hand-side of the corresponding definitions, such terms coincide with the quantities reported above, i.e.

$$C_{\text{tot}}(\Lambda; \mathfrak{e}) = C_{\mathcal{E}}(\Lambda; \mathfrak{e}) \ , \qquad C_{\text{sep,tot}}(\Lambda; \mathfrak{e}) = C_{\text{sep}}(\Lambda; \mathfrak{e}) \ . \quad (48)$$

The same equivalence does not apply to Eqs. (45) and (46): as a matter of fact in these cases, replacing

the ergotropy with the total-ergotropy will lead to quantities that for some channels are provably different from $C_{\text{loc}}(\Lambda; \mathfrak{e})$ and $C_{\text{sep,loc}}(\Lambda; \mathfrak{e})$ – an example of this fact will be provided in App. A 2. Since however the introduction of local capacitances based on a functional that explicitly makes use of global operation makes little sense operationally, in what follows we won't discuss this possibility.

*b. MAWER:–* In the second scenario the number of q-cells composing the quantum battery is treated as a free resource to store a finite energy amount $E$. In the limit in which the number of q-cells is very large compared to the total energy to store (that is, $E/n \to 0$), it's reasonable to expect some advantage in the ratio between input energy and output ergotropy: with the number of cells we are also extending the number of strategies to perform energy injection and extraction, hence we can hope for better efficiencies. In order to evaluate the performance of such schemes we employ the *Maximal Asymptotic Work/Energy Ratio* (MAWER) defined as

$$\mathcal{J}_{\mathcal{E}}(\Lambda) := \limsup_{E \to \infty} \left( \sup_{n \geq 1} \frac{\mathcal{E}^{(n)}(\Lambda; E)}{E} \right) . \qquad (49)$$

It is worth observing that the supremum over $n$ in the above expression can be computed as a $\lim_{n \to \infty}$ (this is a consequence of the monotonicity discussed in the next section). Notice also that, at variance with the capacitances defined in the previous paragraph, the MAWER $\mathcal{J}_{\mathcal{E}}(\Lambda)$ can diverge: explicit examples will be given in Sec. V. Finally, similarly to Eq. (48), one can show that the limit on right-hand-side of Eq. (49) can be evaluated by replacing the optimization of the ergotropy with the total-ergotropy – see Appendix A 3.

## IV. SOME USEFUL PROPERTIES

In the following sections we study the output ergotropy functionals defined in Sec. II for a variety of CPTP channels. To simplify the analysis we'll exploit a series of useful properties that we enlist below.

**Property 0** *[Monotonicity]:–* By construction all our work extraction functionals $\mathcal{W}^{(n)}(\Lambda; E)$ are monotonically increasing both w.r.t. $n$ and $E$, i.e.

$$\mathcal{W}^{(n)}(\Lambda; E) \geq \mathcal{W}^{(n')}(\Lambda; E') , \qquad \forall n \geq n', \quad \forall E \geq E' . \qquad (50)$$

On the contrary, there is no guarantee that for fixed $n$ $\overline{\mathcal{W}}^{(n)}(\Lambda; E)$ is increasing with $E$ (an explicit counter-example for the ergotropy $\mathcal{E}$ is provided in Section V A).

**Property 1** *[Optimality of pure states]:–* In Ref. [36] it was shown that for fixed input energy $E$, the maximum values of the ergotropy, total-ergotropy, and of the non-equilibrium free energy attainable at the output

of any quantum channel can always be achieved by a pure state. In what follows we'll exploit this fact by restricting the maximization in Eq. (25) just to the pure states in $\mathfrak{S}_E^{(n)}$.

**Property 2** *[Super-additivity]:–* As the input energy of the system is an extensive quantity, and local unitary transformations acting on a subset of $n$ quantum cells of a quantum battery define a proper subset of $\mathbb{U}(d^n)$, one can easily show that for both the ergotropy and the total-ergotropy, given any couple of integers $n_1$ and $n_2$, we have

$$\mathcal{W}^{(n_1+n_2)}(\Lambda; E_1 + E_2) \geq \mathcal{W}^{(n_1)}(\Lambda; E_1) + \mathcal{W}^{(n_2)}(\Lambda; E_2) , \qquad (51)$$

for all $E_1 \in [0, n_1]$, $E_2 \in [0, n_2]$, and

$$\mathcal{W}^{(n_1 n_2)}(\Lambda; E) \geq n_2 \mathcal{W}^{(n_1)}(\Lambda; E/n_2) , \qquad (52)$$

for all $E \in [0, n_1 n_2]$.

The fundamental ingredient to prove properties in Eq. (51) and Eq. (52) is the super-addittivity of the ergotropy and total ergotropy for factorized, indepedent systems, i.e. the inequality

$$\mathcal{W}(\hat{\rho}_A \otimes \hat{\rho}_B; \hat{H}_A + \hat{H}_B) \geq \mathcal{W}(\hat{\rho}_A; \hat{H}_A) + \mathcal{W}(\hat{\rho}_B; \hat{H}_B) , \qquad (53)$$

which follows by restricting the maximization over the unitaries acting on the joint system $AB$ to tensor product unitaries acting locally on $A$ and $B$.

To prove Eq. (51) observe then that, given $\hat{\rho}^{(n_1)} \in \mathfrak{S}_{E_1}^{(n_1)}$ and $\hat{\rho}^{(n_2)} \in \mathfrak{S}_{E_2}^{(n_2)}$, we have that $\hat{\rho}^{(n_1)} \otimes \hat{\rho}^{(n_2)} \in \mathfrak{S}_{E_1+E_2}^{(n_1+n_2)}$. Therefore we can write

$$\begin{aligned} \mathcal{W}^{(n_1+n_2)}&(\Lambda; E_1 + E_2) \qquad\qquad\qquad (54) \\ &\geq \mathcal{W}(\Lambda^{\otimes(n_1+n_2)}(\hat{\rho}^{(n_1)} \otimes \hat{\rho}^{(n_2)}); \hat{H}^{(n_1+n_2)}) \\ &= \mathcal{W}(\Lambda^{\otimes(n_1)}(\hat{\rho}^{(n_1)}) \otimes \Lambda^{\otimes n_2}(\hat{\rho}^{(n_2)}); \hat{H}^{(n_1+n_2)}) \\ &\geq \mathcal{W}(\Lambda^{\otimes n_1}(\hat{\rho}^{(n_1)}); \hat{H}^{(n_1)}) + \mathcal{W}(\Lambda^{\otimes n_2}(\hat{\rho}^{(n_2)}); \hat{H}^{(n_2)}) , \end{aligned}$$

where in the second inequality we applied Eq. (53). Taking hence the supremum over all possible choices of $\hat{\rho}^{(n_1)}$ and $\hat{\rho}^{(n_2)}$, we finally arrive to Eq. (51).

The proof of the inequality of Eq. (52) follows a similar path: observe that for $n_1$ integer and $E_1 \in [0, n_1]$, given a generic $\hat{\rho}^{(n_1)} \in \mathfrak{S}_{E_1}^{(n)}$, we have that $(\hat{\rho}^{(n_1)})^{\otimes n_2} \in \mathfrak{S}_{n_2 E_1}^{(n_1 n_2)}$. Accordingly we can write

$$\begin{aligned} \mathcal{W}^{(n_1 n_2)}(\Lambda; n_2 E_1) &\geq \mathcal{W}(\Lambda^{\otimes n_1 n_2}((\hat{\rho}^{(n_1)})^{\otimes n_2}); \hat{H}^{(n_1 n_2)}) \\ &= \mathcal{W}((\Lambda^{\otimes n_1}(\hat{\rho}^{(n_1)}))^{\otimes n_2}; \hat{H}^{(n_1 n_2)}) \\ &\geq n_2 \mathcal{W}((\Lambda^{\otimes n_1}(\hat{\rho}^{(n_1)})); \hat{H}^{(n_1)}) , \quad (55) \end{aligned}$$

where in the third line we invoked once more Eq. (53). Taking now the supremum with respect to all $\hat{\rho}^{(n)} \in \mathfrak{S}_{E_1}^{(n_1)}$ we finally get

$$\mathcal{W}^{(n_1 n_2)}(\Lambda; n_2 E_1) \geq n_2 \mathcal{W}^{(n_1)}(\Lambda; E_1) , \qquad (56)$$

which corresponds to Eq. (52) once we set $E = n_2 E_1$.

It is finally worth stressing that the super-additivity showed by inequalities (51) and (52) also holds for the functionals $\overline{\mathcal{W}}^{(n)}(\Lambda; E)$ and $\overline{\mathcal{W}}_{\text{tot}}^{(n)}(\Lambda; E)$. This is proved by observing that by construction these quantities can be expressed as in Eq. (25) by replacing $\mathfrak{S}_E^{(n)}$ with $\overline{\mathfrak{S}}_E^{(n)}$.

**Property 3** *[Covariance]:–* An important simplification applies for noise models that are well-behaved under unitary transformations that leave the system Hamiltonian invariant, i.e. the transformations identified by the following

**Definition 1.** *A CPTP channel $\Lambda$ is said to be $n$-covariant under energy preserving transformations, if for all $\hat{V}^{(n)} \in \mathbb{U}_{EP}^{(n)}$ there exists $\hat{W}^{(n)} \in \mathbb{U}_{EP}^{(n)}$ such that*

$$\Lambda^{\otimes n} \circ \mathcal{U}_{\hat{V}^{(n)}} = \mathcal{U}_{\hat{W}^{(n)}} \circ \Lambda^{\otimes n} , \qquad (57)$$

*where $\mathcal{U}_{\hat{V}^{(n)}}(\cdot) := \hat{V}^{(n)}(\cdot)\hat{V}^{(n)\dagger}$ is the CPTP map associated with $\hat{V}^{(n)}$, being $\circ$ the composition of super-operators.*

Notice in particular that if $\Lambda$ is $n$-covariant then it's also $m$-covariant for all $m$ integers smaller than $n$, since $\mathbb{U}_{EP}^{(m)}$ induces a proper subgroup in $\mathbb{U}_{EP}^{(n)}$ (on the contrary if $\Lambda$ is $m$-covariant then it is not necessarily $n$-covariant). What is most relevant for us is that, given a $n$-covariant channel $\Lambda$, the maximization in Eq. (25) can be saturated by states that in the eigen-energy basis $\{|E_\ell\rangle\}_\ell$ are represented by a matrix with non-negative terms. To see this recall that from **Property 1** it follows that the maximum in Eq. (25) is achieved by one of the pure states of $\mathfrak{S}_E^{(n)}$. Let hence $\hat{\rho}^{(n)} = |\psi^{(n)}\rangle\langle\psi^{(n)}|$ be one of such states, with $|\psi^{(n)}\rangle = \sum_{\ell=1}^{d^n} \psi_\ell^{(n)}|E_\ell\rangle$. Define now $V^{(n)} \in \mathbb{U}_{EP}^{(n)}$ with phase terms as in Eq. (15) equal to minus the phases of the amplitudes probabilities $\psi_\ell^{(n)}$ of $|\psi^{(n)}\rangle$, i.e. $\phi_\ell := -\arg(\psi_\ell^{(n)})$. By construction the vector

$$|\psi_p^{(n)}\rangle = V^{(n)}|\psi_p^{(n)}\rangle = \sum_{\ell=1}^{d^n} |\psi_\ell^{(n)}||E_\ell\rangle , \qquad (58)$$

is still a pure state of $\mathfrak{S}_E^{(n)}$, whose density operator $\hat{\rho}_p^{(n)} := |\psi^{(n)}\rangle\langle\psi^{(n)}|$ has matrix representation with elements $\langle E_\ell|\hat{\rho}_p^{(n)}|E_{\ell'}\rangle = |\psi_\ell^{(n)}\psi_{\ell'}^{(n)}|$ that are explicitly non-negative. Furthermore, since the channel is $n$-covariant, we can write

$$\begin{aligned}
\mathcal{W}(\Lambda^{\otimes n}(\hat{\rho}_p^{(n)}); \hat{H}^{(n)}) &= \mathcal{W}(\Lambda^{\otimes n} \circ \mathcal{U}_{\hat{V}^{(n)}}(\hat{\rho}^{(n)}); \hat{H}^{(n)}) \\
&= \mathcal{W}(\mathcal{U}_{\hat{W}^{(n)}} \circ \Lambda^{\otimes n}(\hat{\rho}^{(n)}); \hat{H}^{(n)}) \\
&= \mathcal{W}(\Lambda^{\otimes n}(\hat{\rho}^{(n)}); \hat{H}^{(n)}) , \qquad (59)
\end{aligned}$$

where $\hat{W}^{(n)}$ is the element of $\mathbb{U}_{EP}^{(n)}$ defined in Definition 1 and where in the second line we invoked Eq. (16).

**Property 4** *[Ergotropic Equivalence]:–* The analysis of the maximum output ergotropy functional can in part be simplified by the introduction of the following

**Definition 2.** *Two CPTP channels $\Lambda$ and $\Lambda'$ are said to be ergotropically equivalent if*

$$\mathcal{E}^{(n)}(\Lambda'; E) = \mathcal{E}^{(n)}(\Lambda; E) , \quad \forall n, \quad \forall E \in [0, n] . \quad (60)$$

One can easily verify that channels which are ergotropically equivalent are also equivalent in terms of the total ergotropy, and have the same ergotropy capacitances and MAWERs, i.e.

$$C_\mathcal{E}(\Lambda; \mathfrak{e}) = C_\mathcal{E}(\Lambda'; \mathfrak{e}) , \qquad \mathcal{J}_\mathcal{E}(\Lambda) = \mathcal{J}_\mathcal{E}(\Lambda') . \quad (61)$$

A first example of a couple of ergotropically equivalent channels is obtained when $\Lambda$ is a generic transformation and $\Lambda' = \mathcal{U}_{\hat{V}} \circ \Lambda$ for some $\hat{V} \in \mathbb{U}_{EP}^{(1)}$ – the identity in Eq. (60) is then a trivial consequence of the fact that $\hat{V}^{\otimes n}$ is an element of $\mathbb{U}_{EP}^{(n)}$ $(\Lambda')^{\otimes n}$ and of Eq. (16). A slightly more sophisticated example is instead represented by couples of maps where $\Lambda$ is a 1-covariant channel and $\Lambda' = \Lambda \circ \mathcal{U}_{\hat{V}}$ where again $\hat{V} \in \mathbb{U}_{EP}^{(1)}$ – in this case Eq. (60) can be established by using Eq. (57) to express $\Lambda' = \mathcal{U}_{\hat{W}} \circ \Lambda$ to reduce the analysis to the first example.

## V. NOISE MODELS

To test the effectiveness of the new framework described above, here we analyze the energy release efficiency for instances of CPTP maps. We choose two among the fundamental noise models in the landscape of quantum information and communication: the qubit dephasing channel and the depolarizing channel, and we proceed by studying the associated maximum ergotropic functionals defined in the previous sections. The qubit examples explored here were chosen as illustrative cases due to their broad relevance for current quantum computing platforms. However, we emphasize that the formalism itself imposes no restrictions and can be readily applied to more complex quantum battery implementations once the noise processes are characterized.
In the case of the dephasing channel we derive all the quantities defined the previous section and we obtain simple formulas due to the structure of the channel; if the input consists of only two qubits we show that we can find an advantage with entangled input states. For the depolarizing channel we manage to compute the ergotropic capcitances and the MAWER exploiting the symmetries of the channel under unitary rotations; for qudit inputs $(d \geq 3)$ we find that the MAWER goes to infinity, it is a consequence of the fact that in this particular scenario the channel charges the state.

## A. Qubit Dephasing Channel

As a first example we consider the case of quantum batteries made of two-levels quantum systems ($d = 2$) and affected by the detrimental action of a dephasing channel $\Delta_\kappa$ [39, 52] acting on the coherences of the energy eigenvectors of the local Hamiltonian $\hat{h}$, i.e.

$$\Delta_\kappa(|i\rangle\langle i|) \;=\; |i\rangle\langle i|\,, \quad \text{for } i = 0, 1, \tag{62}$$

$$\Delta_\kappa(|0\rangle\langle 1|) \;=\; \sqrt{1-\kappa}\,|0\rangle\langle 1| = \Delta_\kappa^\dagger(|1\rangle\langle 0|)\,, \tag{63}$$

with $\kappa \in [0, 1]$ the dephasing parameter of the model ($\kappa = 1$ corresponding to the complete dephasing).

We start noticing that since such map does not change the mean energy of the input states, so the following upper bound holds:

$$\mathcal{E}^{(n)}(\Delta_\kappa; E) \leq E\,, \qquad \forall n\,, \forall E. \tag{64}$$

Furthermore, since $\Delta_k$ simply rescales the matrix elements of the input state when expressed in the energy eigenbasis, one can easily verify that the channel is $n$-covariant for all $n$. Accordingly in solving the maximization in Eq. (25) we can restrict the analysis to pure states as in Eq. (58), which in the energy eigenbasis have positive amplitude probabilities. If our quantum battery consists of a single q-cell scenario ($n = 1$) for $E \in [0, 1]$ this identifies the vector

$$|\psi_E^{(1)}\rangle := \sqrt{1-E}\,|0\rangle + \sqrt{E}\,|1\rangle\,, \tag{65}$$

as the optimizer state leading to

$$\begin{aligned} \mathcal{E}^{(1)}(\Delta_\kappa; E) \;&=\; \overline{\mathcal{E}}^{(1)}(\Delta_\kappa; E) \\ &=\; E - \frac{1}{2}\left(1 - \sqrt{1 - 4\kappa E(1-E)}\right). \end{aligned} \tag{66}$$

The second identity follows from Eq. (4) and from the fact that $\Delta_\kappa(|\psi_E^{(1)}\rangle\langle\psi_E^{(1)}|)$ admits $\lambda_{1,2} = \frac{1}{2} \pm \frac{1}{2}\sqrt{1 - 4\kappa E(1-E)}$ as eigenvalues; the first identity instead is a consequence of the fact that $\overline{\mathcal{E}}^{(1)}(\Delta_\kappa; E)$ is monotonically increasing for $E \in [0, 1]$ – see First Panel of Fig. 2.

To address the case of a quantum battery comprising an arbitrary number $n$ of q-cells, we observe that given an integer $k \leq n$, the product states of the form $|1\rangle^{\otimes k} \otimes |0\rangle^{n-k}$ are left invariant by $\Lambda^{\otimes n}$: using such inputs we can hence saturate the upper bound of Eq. (64) at least for $E = k$, i.e.

$$\overline{\mathcal{E}}^{(n)}(\Delta_\kappa; E = k) \;=\; \mathcal{E}^{(n)}(\Delta_\kappa; E = k) = k\,, \tag{67}$$

for all $k \in \{0, 1, \cdots, n\}$ – see Fig. 3. For values of $E$ that are not integers the calculation is less immediate and we are not able to provide closed expressions for $\mathcal{E}^{(n)}(\Delta_\kappa; E)$. A numerical study of the special case of $n = 2$ – Fig. 2 Second Panel –reveals that, at variance with what observed in the derivation of Eq. (67), in general the optimal input states will involve some degree

of entanglement. This is explicitly shown in the third panel of Fig. 2 where we report the difference between the value $\overline{\mathcal{E}}^{(2)}(\Delta_\kappa; E)$ obtained by performing an optimization over all possible input states, and the optimal output ergotropy $\overline{\mathcal{E}}^{(2)}_{\text{SEP}}(\Delta_\kappa; E)$ obtained by restricting the maximization over the set of separable input states – see Eq. (21). As evident from the plot a gap can be seen when $\kappa$ is sufficiently large and $E$ is close to the values 0.5 and 1.5. For such choices we have numerical evidences that the optimal state corresponds to the non-factorized vector $|\psi_E^{(2)}\rangle := \sqrt{1 - E/2}\,|00\rangle + \sqrt{E/2}\,|11\rangle$.

### 1. Ergotropic Capacitances and MAWER values

For arbitrary $n$ and $E$, an analytical lower bound for $\mathcal{E}^{(n)}(\Delta_\kappa; E)$ can be obtained by expressing $E$ in terms of its integer part $\lfloor E\rfloor$, and using the super-additivity in Eq. (51) together with Eqs. (67) and (66):

$$\begin{aligned} \mathcal{E}^{(n)}(\Delta_\kappa; E) \;&\geq\; \mathcal{E}^{(n-1)}(\Delta_\kappa; \lfloor E\rfloor) + \mathcal{E}^{(1)}(\Delta_\kappa; \Delta E) \\ &=\; \lfloor E\rfloor + \mathcal{E}^{(1)}(\Delta_\kappa; \Delta E) = E - \delta E\,, \end{aligned} \tag{68}$$

with $\Delta E := E - \lfloor E\rfloor \in [0, 1]$ and

$$\delta E := \frac{1}{2}(1 - \sqrt{1 - 4\kappa\Delta E(1 - \Delta E)}) \in [0, \Delta E]\,.$$

Observing that by construction $\delta E$ is smaller than 1, from the above inequality and from Eq. (64) we get the following inequalities

$$\mathfrak{e} \geq \frac{\mathcal{E}^{(n)}(\Delta_\kappa; n\mathfrak{e})}{n} \geq \mathfrak{e} - 1/n\,, \tag{69}$$

$$1 \geq \frac{\mathcal{E}^{(n)}(\Delta_\kappa; E)}{E} \geq 1 - 1/E\,, \tag{70}$$

which finally translate to closed expressions for the ergotropic capacitance and for the MAWER of the model

$$C_\mathcal{E}(\Delta_\kappa; \mathfrak{e}) \;=\; \mathfrak{e}\,, \qquad \forall \mathfrak{e} \in [0, 1]\,, \tag{71}$$

$$\mathcal{J}_\mathcal{E}(\Delta_\kappa) \;=\; 1\,. \tag{72}$$

We remark that, as the lower bound in Eq. (68) is attainable using separable states of the form

$$|1\rangle^{\otimes\lfloor E\rfloor} \otimes |0\rangle^{\otimes(n-\lfloor E\rfloor-1)} \otimes |\psi_{\mathfrak{e}'}^{(1)}\rangle, \tag{73}$$

here $|\psi_{\mathfrak{e}'}^{(1)}\rangle$ is defined as in Eq. 65 and $\mathfrak{e}' = E - \lfloor E\rfloor$; despite the advantages reported in the previous section in the finite $n$ regime, the asymptotic values of Eq. (69) and Eq. (70) can be obtained without the explicit use of entangled input states. Furthermore as the energy recovery from the input state in Eq. (73) only requires local operations, for the dephasing channels we have that all the channel capacitances coincide leading to a collapse of the hierarchy of inequalities in Eq. (47), i.e.

$$C_\mathcal{E}(\Delta_\kappa; \mathfrak{e}) = C_{\text{loc}}(\Delta_\kappa; \mathfrak{e}) = C_{\text{sep}}(\Delta_\kappa; \mathfrak{e}) = C_{\text{sep,loc}}(\Delta_\kappa; \mathfrak{e})\,. \tag{74}$$

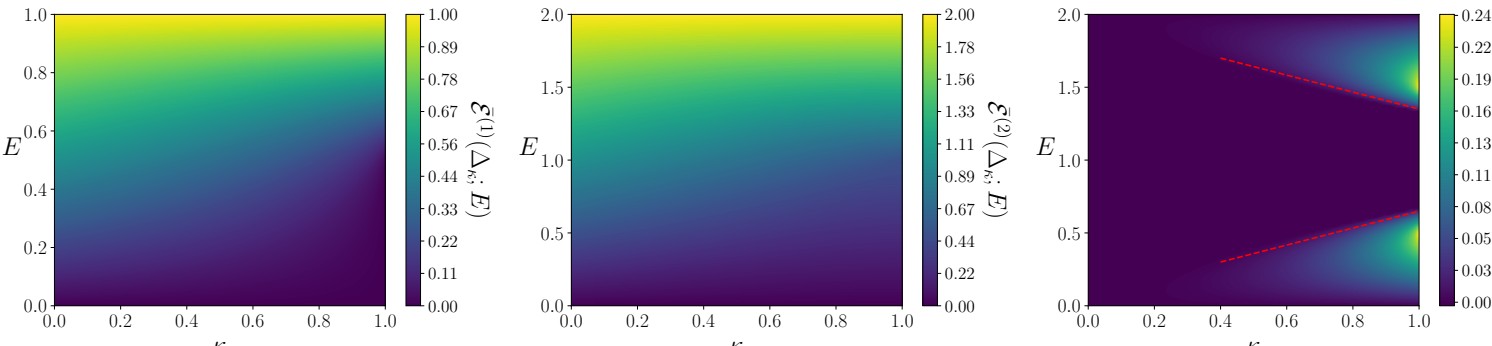

FIG. 2. Study of the output maximum ergotropy per site $\overline{\mathcal{E}}^{(n)}(\Delta_\kappa; E)$ for two-level quantum batteries evolving under the action of a dephasing channel $\Delta_\kappa$. First panel Single site scenario ($n = 1$): in this case $\overline{\mathcal{E}}^{(1)}(\Delta_\kappa; E)$ is given in Eq. (66). Notice that for $E = 1$ (i.e. when no energy constraint is imposed on the input state of the quantum battery), the function saturates to the maximum stored energy value, i.e. $\overline{\mathcal{E}}^{(1)}(\Delta_\kappa; 1) = 1$; Second panel two-sites scenario ($n = 2$). Here the optimal value of $\overline{\mathcal{E}}^{(2)}(\Delta_\kappa; E)$ is obtained solving the optimization numerically. As in the case $n = 1$ case, the quantity appears to be monotonically increasing in $E$, indicating that $\overline{\mathcal{E}}^{(2)}(\Delta_\kappa; E) = \mathcal{E}^{(2)}(\Delta_\kappa; E)$. Third panel Entanglement boost: difference between $\overline{\mathcal{E}}^{(2)}(\Delta_\kappa; E)$ and the optimal output ergotropy $\overline{\mathcal{E}}^{(2)}_{\mathrm{SEP}}(\Delta_\kappa; E)$ obtained by restricting the optimization over the set of separable input states. The red dashed lines highlight the area in the parameter space where the entangled input states have an advantage over the separable input strategy.

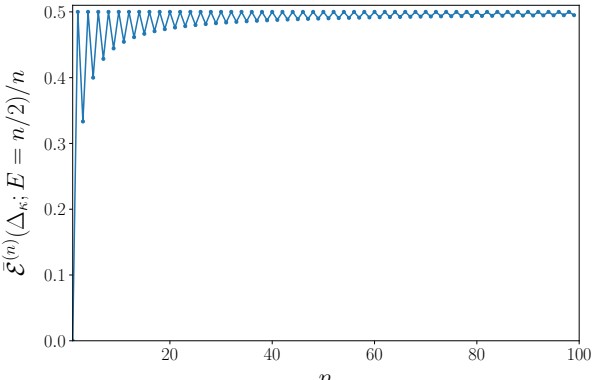

FIG. 3. Maximal output ergotropy per q-cell copies $\overline{\mathcal{E}}^{(n)}(\Delta_\kappa; E = n/2)/n$ at the output of the completely dephasing channel $\Delta_{\kappa=1}$ as a function of $n$: for $n$ even this quantity saturates to the maximum 1 (see Eq. (67)); for even $n$ the plotted values are the result of a numerical optimization.

This collapse of the hierarchy reveals a key physical property of the dephasing noise: the stability of the stored energy against dephasing is not enhanced by quantum correlations across cells or non-local control at the level of work extraction operations.

For what concerns the free energy we find a relation

that is analogous to Eq. (68)

$$\mathcal{F}^{(n)}_\beta(\Delta_\kappa; E) \geq \mathcal{F}^{(n-1)}_\beta(\Delta_\kappa; \lfloor E \rfloor) + \mathcal{F}^{(1)}_\beta(\Delta_\kappa; \Delta E)$$
$$= \lfloor E \rfloor + \mathcal{F}^{(1)}_\beta(\Delta_\kappa; \Delta E) . \quad (75)$$

The lower bound in Eq. (75) is attainable with the state in Eq. (73) so we have that

$$\mathfrak{e} \geq \frac{\mathcal{F}^{(n)}_\beta(\Delta_\kappa; n\mathfrak{e})}{n} \geq \mathfrak{e} - 1/n , \quad (76)$$

due to the above inequality we finally obtain the following expression for the free energy capacitance:

$$C_\beta(\Delta_\kappa; \mathfrak{e}) = \mathfrak{e} + \frac{\log Z_\beta(\hat{h})}{\beta} . \quad (77)$$

As expected, the presence of an additional resource - the heat bath - allows for the retrieval of more work than what would be possible to extract without it, in a fully analogous way to what happens for noiseless work extraction processes [46].

### 2. Dephasing channels in higher dimensions

The results reported above can be generalized to noisy quantum batteries composed by q-cells of arbitrary dimension $d > 2$, irrespective of the spectrum of the local Hamiltonian $\hat{h}$ and of the specific structure of the dephasing coefficients one can assign to the various off-diagonal terms. In particular, identifying with $|0\rangle$ and $|1\rangle$ the

ground and maximal energy state of a single q-cell, one can use the state in Eq. (73) to show that the bounds of Eqs. (69) and (70) still apply, from which Eqs. (72) and (74) can then be easily recovered.

## B. Depolarizing Channel

The depolarizing channel is one of the simplest and most studied - because of its symmetries - noise models in quantum information theory: specifically, it is co-variant w.r.t. the action of any unitary transformation [52, 53]. In the qubit setting it describes, depending on a probability parameter, the simultaneous action of bit-flip, phase-flip and bit-phase-flip errors. More generally for a qudit system the depolarizing channel $\mathcal{D}_\lambda$ induces a mapping that can be expressed by the following:

$$\mathcal{D}_\lambda(\hat{\rho}) := \lambda\hat{\rho} + (1-\lambda)\mathrm{Tr}[\hat{\rho}]\,\frac{\hat{\mathbb{1}}}{d}\,, \qquad (78)$$

where $\hat{\mathbb{1}}$ is the identity operator and $\lambda \in [-1/(d^2-1), 1]$ is a noise parameter that characterizes the transformation. In particular for $\lambda \in [0, 1]$ the map represents an incoherent mixture between the input state and the completely mixed state of the model; while for $\lambda \in [-1/(d^2-1), 0)$ it induces an inversion with respect to the identity operator: in the qubit case for instance it results in a universal NOT in the Bloch sphere combined with a contraction of the Bloch vector [54].

Simple algebra reveals that $\mathcal{D}_\lambda$ induces a linear transformation of the input energy of the system. Specifically being $\hat{\rho}^{(n)} \in \mathfrak{S}_E^{(n)}$ an input state of energy $E$, we have that its output $\mathcal{D}_\lambda^{\otimes n}(\hat{\rho}^{(n)})$ is an element of $\mathfrak{S}_{E'}^{(n)}$ with

$$E' := \lambda E + (1-\lambda)n\frac{\mathrm{Tr}[\hat{h}]}{d}\,, \qquad (79)$$

that allows us to replace Eq. (64) with

$$\mathcal{E}^{(n)}(\mathcal{D}_\lambda; E) \leq \lambda E + (1-\lambda)n\frac{\mathrm{Tr}[\hat{h}]}{d}\,, \quad \forall n\,, \forall E\,. \quad (80)$$

We notice also that, as in the case of the dephasing channel $\Delta_\kappa$, also $\mathcal{D}_\lambda$ is $n$-covariant so that we can always restrict the optimization in Eq. (27) to pure vectors of the form of the one in Eq. (58). As shown in App. B this implies that, in the case of a quantum battery with a single q-cell ($n = 1$), irrespective of the dimensionality of the model, the maximum output ergotropy at energy less or equal than $E$ is attained by pure states with average energy $E$, so it is given by

$$\mathcal{E}^{(1)}(\mathcal{D}_\lambda; E) = \overline{\mathcal{E}}^{(1)}(\mathcal{D}_\lambda; E) = \lambda E + D(\lambda)\,, \qquad (81)$$

with $D(\lambda)$ a discontinuous function being equal to 0 for $\lambda \geq 0$, assuming instead the value $-\lambda$ for $\lambda < 0$. Accordingly we get

$$\mathcal{E}^{(1)}(\mathcal{D}_\lambda; E) = \begin{cases} \overline{\mathcal{E}}^{(1)}(\mathcal{D}_\lambda; E) = \lambda E & \text{for } \lambda \geq 0, \\ \overline{\mathcal{E}}^{(1)}(\mathcal{D}_\lambda; 0) = |\lambda| & \text{for } \lambda \leq 0, \end{cases} \quad (82)$$

where we exploit the fact that the expression in Eq. (81) is non-decreasing (non-increasing) w.r.t. $E$ for positive (negative) $\lambda$ values – see top panels of Fig. 4. In a similar way we get

$$\overline{\mathcal{E}}_{\mathrm{tot}}^{(1)}(\mathcal{D}_\lambda; E) = \lambda E + D_{\mathrm{tot}}(\lambda; \hat{h})\,, \qquad (83)$$

$$\mathcal{E}_{\mathrm{tot}}^{(1)}(\mathcal{D}_\lambda; E) = \begin{cases} \lambda E + D_{\mathrm{tot}}(\lambda; \hat{h}) & \text{for } \lambda \geq 0, \\ D_{\mathrm{tot}}(\lambda; \hat{h}) & \text{for } \lambda \leq 0, \end{cases} \quad (84)$$

where $D_{\mathrm{tot}}(\lambda; \hat{h})$ is a constant term only depending on the spectrum of the single-site Hamiltonian $\hat{h}$ and on the noise parameter $\lambda$ – see top panels of Fig. 4 and App. B for details. The simple form of $\mathcal{E}^{(1)}(\mathcal{D}_\lambda; E)$ and $\mathcal{E}^{(1)}(\mathcal{E}_\lambda; E)$ is a consequence both of the symmetry of the depolarizing channel, since it commutes with any unitary transformation, and of the equation (79), which tells us that the output energy is just a function of solely the input energy of the state. We stress that also in this case the maximum for $\overline{\mathcal{E}}_{\mathrm{tot}}^{(1)}(\mathcal{D}_\lambda; E)$ is attained by any pure input state that fulfils the energy constraint. We notice also that, while non evident by the formula, in the special case of $d = 2$ (qubit) $D_{\mathrm{tot}}(\lambda; \hat{h})$ corresponds to $D(\lambda)$ and Eqs. (83) and (84) reduce to Eqs. (81) and (82).

### 1. Ergotropic Capacitances

Also for this channel we are able to compute the capacitances and the MAWERs of the model. The key argument follows from the result of King presented in Ref. [53] which shows that the minimal entropy at the output of a multi-use depolarizing channel is additive, i.e. that for any $n$ and for any given $\rho^{(n)}$ state of $n$ q-cells one has

$$\begin{aligned} S(\mathcal{D}_\lambda^{\otimes n}(\rho^{(n)})) &\geq S((\mathcal{D}_\lambda(|\psi\rangle\langle\psi|))^{\otimes n}) \\ &= nS(\mathcal{D}_\lambda(|\psi\rangle\langle\psi|)) = nS_d(\lambda)\,, \quad (85) \end{aligned}$$

with $|\psi\rangle$ a generic pure input state of a single q-cell, and $S_d(\lambda)$ being the associated output entropy (see Eq. (B7) of App. B). Notice that due to the special symmetry of $\mathcal{D}_\lambda$, the specific choice of $|\psi\rangle$ doesn't matter: in particular, if $\rho^{(n)}$ belongs to the energy constrained subset $\overline{\mathfrak{S}}_E^{(n)}$ by identifying $|\psi\rangle$ with the vector $|\psi_{E/n}\rangle$ which has mean energy $E/n$, we can ensure that also $|\psi_{E/n}\rangle\langle\psi_{E/n}|^{\otimes n}$ belongs to the same set. Accordingly for all $n \in \mathbb{N}$ and $E \in [0, n]$ we have

$$\min_{\hat{\rho}^{(n)} \in \overline{\mathfrak{S}}_E^{(n)}} \frac{S(\mathcal{D}_\lambda^{\otimes n}(\hat{\rho}^{(n)}); \hat{H}^{(n)})}{n} = S_d(\lambda)\,, \qquad (86)$$

which, exploiting the monotonicity of the total ergotropy as in Eq. (10), leads to

$$\begin{aligned} \max_{\hat{\rho}^{(n)} \in \overline{\mathfrak{S}}_E^{(n)}} \frac{\mathcal{E}_{\mathrm{tot}}(\mathcal{D}_\lambda^{\otimes n}(\hat{\rho}^{(n)}); \hat{H}^{(n)})}{n} &= \mathcal{E}_{\mathrm{tot}}(\mathcal{D}_\lambda(|\psi_{E/n}\rangle\langle\psi_{E/n}|)) \\ &= \overline{\mathcal{E}}_{\mathrm{tot}}^{(1)}(\mathcal{D}_\lambda; E/n)\,, \quad (87) \end{aligned}$$

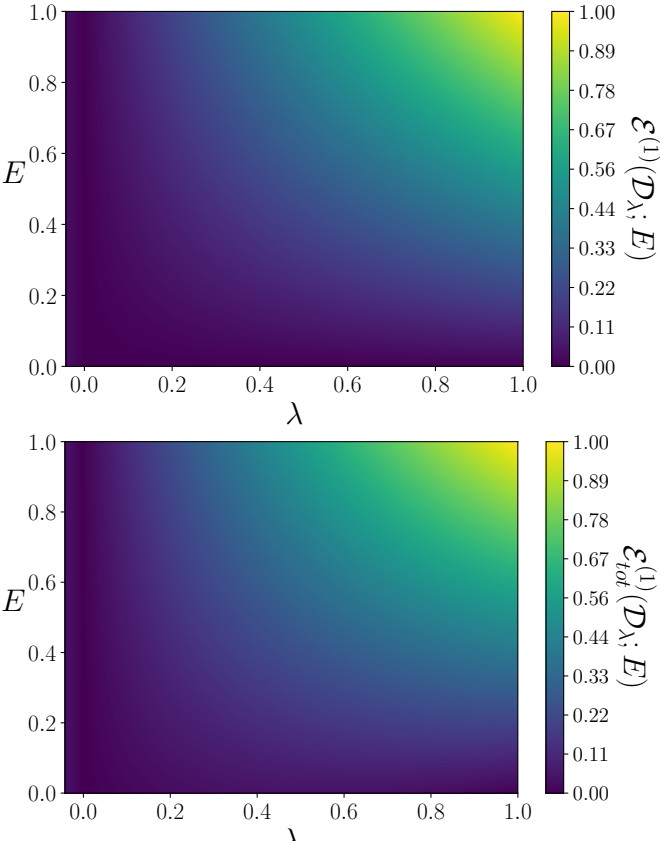

FIG. 4. Upper panel: plot of the single site, maximum output ergotropy $\mathcal{E}^{(1)}(\mathcal{D}_\lambda; E)$ for the depolarizing channel (see Eq. (82)). As shown in Eq. (96) the plot represents the local egotropy capacitance $C_{\text{loc}}(\Lambda; \mathfrak{e})$ and the separable local ergotropy capacitance $C_{\text{sep,loc}}(\Lambda; \mathfrak{e})$. Lower panel: plot of the single site, maximum output total-ergotropy functional $\mathcal{E}^{(1)}_{\text{tot}}(\mathcal{D}_\lambda; E)$ for the case of q-cells of dimension $d = 5$ with non-degenerate, equally spaced energy spectra (see Eq. (84)). As shown in Eq. (90) the plot on the right represents the egotropy capacitance $C_{\mathcal{E}}(\Lambda; \mathfrak{e})$ and the separable-input ergotropy capacitance $C_{\text{sep}}(\Lambda; \mathfrak{e})$.

and

$$\max_{\hat{\rho}^{(n)} \in \mathfrak{S}^{(n)}_E} \frac{\mathcal{E}_{\text{tot}}(\mathcal{D}^{\otimes n}_\lambda(\hat{\rho}^{(n)}); \hat{H}^{(n)})}{n} = \mathcal{E}^{(1)}_{\text{tot}}(\mathcal{D}_\lambda; E/n) \ , \quad (88)$$

with $\overline{\mathcal{E}}^{(1)}_{\text{tot}}(\mathcal{D}_\lambda; E/n)$ and $\mathcal{E}^{(1)}_{\text{tot}}(\mathcal{D}_\lambda; E/n)$ defined as in Eqs. (83) and (84). In a similar fashion, exploiting the fact that $|\psi_{E/n}\rangle\langle\psi_{E/n}|^{\otimes n}$ is separable we can also write

$$\max_{\hat{\rho}^{(n)}_{\text{sep}} \in \mathfrak{S}^{(n)}_E} \frac{\mathcal{E}_{\text{tot}}(\mathcal{D}^{\otimes n}_\lambda(\hat{\rho}^{(n)}); \hat{H}^{(n)})}{n} = \mathcal{E}^{(1)}_{\text{tot}}(\mathcal{D}_\lambda; E/n) \ . \quad (89)$$

Replacing Eqs. (88) and (89) in the right-hand-side of Eqs. (A8) and (A9) and using the identities in Eq. (48) we can then conclude that

$$C_{\mathcal{E}}(\mathcal{D}_\lambda; \mathfrak{e}) = C_{\text{sep}}(\mathcal{D}_\lambda; \mathfrak{e}) = \mathcal{E}^{(1)}_{\text{tot}}(\mathcal{D}_\lambda; \mathfrak{e}) \ . \quad (90)$$

Consider next the case of $C_{\text{loc}}(\mathcal{D}_\lambda; \mathfrak{e})$ and $C_{\text{sep,loc}}(\mathcal{D}_\lambda; \mathfrak{e})$. Observe that for a generic $\hat{\rho}^{(n)} \in \overline{\mathfrak{S}}^{(n)}_E$ we can write

$$\mathcal{E}_{\text{loc}}(\mathcal{D}^{\otimes n}_\lambda(\hat{\rho}^{(n)}); \hat{H}^{(n)}) = \sum_{j=1}^n \mathcal{E}(\mathcal{D}_\lambda(\hat{\rho}^{(n)}_j); \hat{h}) \ , \quad (91)$$

where given $j \in \{1, \cdots, n\}$, $\hat{\rho}^{(n)}_j$ is the reduced density matrix of $\hat{\rho}^{(n)}$ associated with the $j$-th q-cell, whose input energies $E_j = \mathfrak{C}(\hat{\rho}^{(n)}_j; \hat{h})$ fulfil $\sum_{j=1}^n E_j = E$. Invoking **Property 1** we can now write

$$\mathcal{E}(\mathcal{D}_\lambda(\hat{\rho}^{(n)}_j); \hat{h}) \leq \mathcal{E}(\mathcal{D}_\lambda(|\psi_{E_j}\rangle\langle\psi_{E_j}|); \hat{h}) = \overline{\mathcal{E}}^{(1)}(\mathcal{D}_\lambda; E_j) \ , \quad (92)$$

with $|\psi_{E_j}\rangle$ a single-site pure state with input energy that exactly matches $E_j$ and with $\overline{\mathcal{E}}^{(1)}(\mathcal{D}_\lambda; E_j)$ the optimal value given by Eq. (81). Replacing Eq. (92) into Eq. (91) we can hence write

$$\mathcal{E}_{\text{loc}}(\mathcal{D}^{\otimes n}_\lambda(\hat{\rho}^{(n)}); \hat{H}^{(n)}) \leq \sum_{j=1}^n \overline{\mathcal{E}}^{(1)}(\mathcal{D}_\lambda; E_j)$$
$$= n\overline{\mathcal{E}}^{(1)}(\mathcal{D}_\lambda; E/n) \ , \quad (93)$$

where in the last step we used the functional dependence of $\overline{\mathcal{E}}^{(1)}(\mathcal{D}_\lambda; E)$ upon $E$ as expressed by Eqs. (83) and (84). Notice that the upper bound is a value that $\mathcal{E}_{\text{loc}}(\mathcal{D}^{\otimes n}_\lambda(\hat{\rho}^{(n)}); \hat{H}^{(n)})$ can attain on $\overline{\mathfrak{S}}^{(n)}_E$ by using as input the separable state $|\psi_{E_1}\rangle \otimes |\psi_{E_2}\rangle \otimes \cdots \otimes |\psi_{E_n}\rangle$. Accordingly we can write

$$\max_{\hat{\rho}^{(n)}_{\text{sep}} \in \overline{\mathfrak{S}}^{(n)}_E} \frac{\mathcal{E}_{\text{loc}}(\mathcal{D}^{\otimes n}_\lambda(\hat{\rho}^{(n)}); \hat{H}^{(n)})}{n} = \overline{\mathcal{E}}^{(1)}(\mathcal{D}_\lambda; E/n) \ , \quad (94)$$

and hence

$$\max_{\hat{\rho}^{(n)}_{\text{sep}} \in \mathfrak{S}^{(n)}_E} \frac{\mathcal{E}_{\text{loc}}(\mathcal{D}^{\otimes n}_\lambda(\hat{\rho}^{(n)}); \hat{H}^{(n)})}{n} = \mathcal{E}^{(1)}(\mathcal{D}_\lambda; E/n) \ , \quad (95)$$

which inserted in Eqs. (45) and (46) leads to

$$C_{\text{loc}}(\mathcal{D}_\lambda; \mathfrak{e}) = C_{\text{sep,loc}}(\mathcal{D}_\lambda; \mathfrak{e}) = \mathcal{E}^{(1)}(\mathcal{D}_\lambda; \mathfrak{e}) \ . \quad (96)$$

Equations (90) and (96) show that for $d > 2$, in the case of depolarizing channels, the use of entangled states does not improve the energy extraction process, while the possibility of using global operations can provide an advantage.

An analogous result can be found in the case of the free energy. The depolarizing channel is covariant under any unitary transformation and thanks to Eq. (85) we can show that

$$\frac{\mathcal{F}^{(n)}_\beta(\mathcal{D}_\lambda; n\mathfrak{e})}{n} = \mathcal{F}^{(1)}_\beta(\mathcal{D}_\lambda; \mathfrak{e})$$
$$= \lambda\mathfrak{e} + \frac{(1-\lambda)}{d} \text{Tr}\left[\hat{h}\right] - S_d(\lambda) \ , \quad (97)$$

so the corresponding capacitance can be evaluated as

$$C_\beta(\mathcal{D}_\lambda; \mathfrak{e}) = \mathcal{F}_\beta^{(1)}(\mathcal{D}_\lambda; \mathfrak{e}) + \frac{\log Z_\beta(\hat{h})}{\beta} . \qquad (98)$$

This result shows that also in the presence of a thermal bath the energy release process is not boosted by input entangled states.

### 2. MAWER values

To evaluate the MAWER for the depolarizing channel we can use the identities in Eqs. (A25), (88) and (84). For $\lambda < 0$ these tell us that $\mathcal{J}_\mathcal{E}(\mathcal{D}_\lambda)$ is unbounded, i.e.

$$\mathcal{J}_\mathcal{E}(\mathcal{D}_\lambda) = \limsup_{E \to \infty} \left( \sup_{n \geq 1} \frac{n D_{\text{tot}}(\lambda; \hat{h})}{E} \right) = \infty . \qquad (99)$$

For $\lambda \geq 0$ we need to distinguish the case $d = 2$ from the rest. If $d > 2$, since $D_{\text{tot}}(\lambda; \hat{h}) > 0$ we get that once again $\mathcal{J}_\mathcal{E}(\mathcal{D}_\lambda)$ diverges

$$\mathcal{J}_\mathcal{E}(\mathcal{D}_\lambda) = \limsup_{E \to \infty} \left( \sup_{n \geq 1} \frac{n[E/n + D_{\text{tot}}(\lambda; \hat{h})]}{E} \right) = \infty . \tag{100}$$

For $d = 2$ on the contrary since $D_{\text{tot}}(\lambda; \hat{h}) = D(\lambda) = 0$, we get

$$\mathcal{J}_\mathcal{E}(\mathcal{D}_\lambda) = \limsup_{E \to \infty} \left( \sup_{n \geq 1} \frac{n E/n}{E} \right) = \lambda . \qquad (101)$$

## VI. CONCLUSIONS

In this work we have analyzed the efficiency of work extraction for Quantum Batteries formed by a collection of identical and independent quantum cells (each one of them being described by a noisy quantum system). For this purpose, we have introduced a theoretical framework that allows us to understand the action of quantum effects in the noisy energy storage scenario. Specifically, we have defined the work capacitance and maximal asymptotic work/energy ratio (MAWER) to characterize the scalable work output and efficiency of quantum batteries operating under noise. These figures of merit provide an operationally meaningful assessment of stability and robustness against noise during cyclic charging and discharging protocols. As opposed to prior approaches focused on charging power on short timescales, our formalism reveals the possible long-term benefits of quantum battery designs in the presence of noise.

We have been able to identify some general properties of the optimal initial state both for the work capacitances and the MAWERs. We have applied our methods to two instances of relevant noise models (dephasing and depolarizing noise), in both of these situations we have been able to identify the optimal initial state. In this special setting we managed to show that in the limit of quantum batteries composed by infinite q-cells the most prominent role is played by the power of global operations, while for a fixed number of q-cells input entanglement can be beneficial, as in the case of the dephasing channel.

We acknowledge financial support by MIUR (Ministero dell' Istruzione, dell' Universitá e della Ricerca) by PRIN 2017 Taming complexity via Quantum Strategies: a Hybrid Integrated Photonic approach (QUSHIP) Id. 2017SRN-BRK, and via project PRO3 Quantum Pathfinder. S.C. is also supported by a grant through the IBM-Illinois Discovery Accelerator Institute.

## Appendix A: Capacitance and MAWER characterization

In Sec. A 1 we prove that the lim sup in Eqs. (43) and (A8) correspond to regular limits. In Sec. A 2 derive the identity stated in Eq. (48). Finally in Sec. A 3 we show that in the definition of the MAWER the limit on the right-hand-side of Eq. (49) can be evaluated by replacing the optimization of the ergotropy functional with the total-ergotropy.

### 1. Existence of the ergotropic capacitance

In this section we show that for both the ergotropy and the total-ergoropy functions, the limit for $n \to \infty$

$$w_\mathfrak{e}^{(n)} := \frac{\mathcal{W}^{(n)}(\Lambda; E = n\mathfrak{e})}{n} , \qquad (A1)$$

exists finite for all $\mathfrak{e} \in [0, 1]$, and that it corresponds to the maximum value these functionals assume with respect to $n$. This implies that the lim sup in both Eqs. (43) and (A8) can be replaced with the limit, yielding the identities

$$C_\mathcal{W}(\Lambda; \mathfrak{e}) = \lim_{n \to \infty} w_\mathfrak{e}^{(n)}(\Lambda) = \sup_{n \geq 1} w_\mathfrak{e}^{(n)}(\Lambda) . \qquad (A2)$$

The fundamental tool to prove Eq. (A2) is the weakly-increasing property which we define as follows:

**Definition 3.** *A real function $f_n$ on the set of integer number, is said to be Weakly-Increasing (W-I) if the following properties hold true:*

$$f_{nk} \geq f_n , \qquad (A3)$$

$$f_{n+k} \geq \frac{n}{n+k} f_n + \frac{k}{n+k} f_k , \qquad (A4)$$

*for all $n$ and $k$ integers.*

Observe that in the case of $w_\mathfrak{e}^{(n)}(\Lambda)$, Eqs. (A3) and (A4) hold as direct consequences of Eqs. (52) and (51)

respectively. Indeed we can write

$$w_{\mathfrak{e}}^{(nk)}(\Lambda) = \frac{\mathcal{W}^{(nk)}(\Lambda; nk\mathfrak{e})}{nk} \geq \frac{\mathcal{W}^{(n)}(\Lambda; n\mathfrak{e})}{n} = w_{\mathfrak{e}}^{(n)}(\Lambda) , \tag{A5}$$

and

$$\begin{aligned} w_{\mathfrak{e}}^{(n+k)}(\Lambda) &= \frac{\mathcal{W}^{(n+k)}(\Lambda; (n+k)\mathfrak{e})}{n+k} \\ &\geq \frac{\mathcal{W}^{(n)}(\Lambda; n\mathfrak{e}) + \mathcal{W}^{(k)}(\Lambda; k\mathfrak{e})}{n+k} \\ &= \frac{n}{n+k} w_{\mathfrak{e}}^{(n)}(\Lambda) + \frac{k}{n+k} w_{\mathfrak{e}}^{(k)}(\Lambda) . \end{aligned} \tag{A6}$$

Now, simple algebraic considerations show that any limited W-I function $f_n$ admits (finite) limit as $n$ goes to infinity, and that such limit corresponds to its supremum value with respect to $n$, i.e.

**Lemma 1.** *Let $f_n$ be a real function defined on the integers which is finite. If $f_n$ is W-I then we have*

$$\lim_{n\to\infty} f_n = \sup_{n\geq 1} f_n , \tag{A7}$$

*(the existence of the supremum being ensured by the finiteness of $f_n$).*

Since $w_{\mathfrak{e}}^{(n)}(\Lambda)$ is finite by definition (it assumes values on the interval $[0,1]$), Eq. (A2) follows as a consequence of Lemma 1.

### 2. Equivalence between ergotropic capacitances and total-ergotropy capacitances

Here we prove Eq. (48) which implies that the total-ergotropy capacitance and the separable-input ergotropic capacitance defined as

$$C_{\text{tot}}(\Lambda; \mathfrak{e}) := \limsup_{n\to\infty} \max_{\hat{\rho}^{(n)}\in\mathfrak{S}_{E=n\mathfrak{e}}^{(n)}} \frac{\mathcal{E}_{\text{tot}}(\Lambda^{\otimes n}(\hat{\rho}^{(n)}); \hat{H}^{(n)})}{n} , \tag{A8}$$

$$C_{\text{sep,tot}}(\Lambda; \mathfrak{e}) := \limsup_{n\to\infty} \max_{\hat{\rho}_{\text{sep}}^{(n)}\in\mathfrak{S}_{E=n\mathfrak{e}}^{(n)}} \frac{\mathcal{E}_{\text{tot}}(\Lambda^{\otimes n}(\hat{\rho}_{\text{sep}}^{(n)}); \hat{H}^{(n)})}{n} , \tag{A9}$$

coincide with the original values given in Eqs. (43) and (44) respectively.

Let's start by observing that since the total-ergotropy of a state provides a natural upper bound for its ergotropy we have that $C_{\text{tot}}(\Lambda; \mathfrak{e})$ and $C_{\text{sep,tot}}(\Lambda; \mathfrak{e})$ are always greater than or equal to $C(\Lambda; \mathfrak{e})$ and $C_{\text{tot}}(\Lambda; \mathfrak{e})$ respectively, i.e.

$$C_{\text{tot}}(\Lambda; \mathfrak{e}) \geq C_{\mathcal{E}}(\Lambda; \mathfrak{e}) , \qquad C_{\text{sep,tot}}(\Lambda; \mathfrak{e}) \geq C_{\text{sep}}(\Lambda; \mathfrak{e}) . \tag{A10}$$

In order to prove the identities in Eq. (48) it's hence sufficient to verify that the reverse inequalities are also valid. To show this observe that, given $n$ integer and $\hat{\rho}^{(n)} \in \mathfrak{S}_{E=n\mathfrak{e}}^{(n)}$, we can write

$$\begin{aligned} \frac{\mathcal{E}_{\text{tot}}(\Lambda^{\otimes n}(\hat{\rho}^{(n)}); \hat{H}^{(n)})}{n} &= \lim_{N\to\infty} \frac{\mathcal{E}((\Lambda^{\otimes n}(\hat{\rho}^{(n)}))^{\otimes N}; \hat{H}^{(nN)})}{nN} \\ &= \lim_{N\to\infty} \frac{\mathcal{E}(\Lambda^{\otimes nN}((\hat{\rho}^{(n)})^{\otimes N}); \hat{H}^{(nN)})}{nN} \\ &\leq \lim_{N\to\infty} \max_{\hat{\rho}^{(nN)}\in\mathfrak{S}_{E=nN\mathfrak{e}}^{(nN)}} \frac{\mathcal{E}(\Lambda^{\otimes nN}((\hat{\rho}^{(nN)})); \hat{H}^{(nN)})}{nN} \\ &\leq \lim_{N\to\infty} \frac{\mathcal{E}^{(nN)}(\Lambda; nNE)}{nN} = C_{\mathcal{E}}(\Lambda; \mathfrak{e}) , \end{aligned} \tag{A11}$$

where in the third passage we used the fact that $(\hat{\rho}^{(n)})^{\otimes N} \in \mathfrak{S}_{E=nN\mathfrak{e}}^{(nN)}$. Taking hence the supremum over all $\hat{\rho}^{(n)} \in \mathfrak{S}_{E=n\mathfrak{e}}^{(n)}$ and then taking the limit $n \to \infty$ we get

$$C_{\text{tot}}(\Lambda; \mathfrak{e}) \leq C_{\mathcal{E}}(\Lambda; \mathfrak{e}) , \tag{A12}$$

which, together with Eq. (A10), proves the first of the identities of Eq. (48). The proof of the second one follows in a similar way noticing that, given $\hat{\rho}_{\text{sep}}^{(n)}$ a separable state of $\mathfrak{S}_{E=n\mathfrak{e}}^{(n)}$, one has that $(\hat{\rho}_{\text{sep}}^{(n)})^{\otimes N}$ is a separable state of $\mathfrak{S}_{E=nN\mathfrak{e}}^{(nN)}$: this allows us to replicate the steps in Eq. (A11) obtaining

$$\frac{\mathcal{E}_{\text{tot}}(\Lambda^{\otimes n}(\hat{\rho}_{\text{sep}}^{(n)}); \hat{H}^{(n)})}{n} \leq C_{\text{sep}}(\Lambda; \mathfrak{e}) . \tag{A13}$$

Taking then the max over all possible $\hat{\rho}_{\text{sep}}^{(n)} \in \mathfrak{S}_{E=n\mathfrak{e}}^{(n)}$ and sending $n \to \infty$ we hence get

$$C_{\text{sep,tot}}(\Lambda; \mathfrak{e}) \leq C_{\text{sep}}(\Lambda; \mathfrak{e}) , \tag{A14}$$

which together with Eq. (A10) gives the second identity of Eq. (48).

We now prove that the equivalent of Eq. (48) does not hold for the local total ergotropy capacitance for separable inputs. In other words we show that replacing $\mathcal{E}$ with $\mathcal{E}_{\text{tot}}$ in the right-hand-side of Eqs. (45) and (46) will in general lead to quantities which are different from $C_{\text{loc}}(\Lambda; \mathfrak{e})$ and $C_{\text{sep,loc}}(\Lambda; \mathfrak{e})$ respectively. We can name these new objects the *local total-ergotropy capacitance*

$$C_{\text{loc,tot}}(\Lambda; \mathfrak{e}) \tag{A15}$$
$$:= \limsup_{n\to\infty} \max_{\hat{\rho}^{(n)}\in\mathfrak{S}_{E=n\mathfrak{e}}^{(n)}} \frac{\mathcal{E}_{\text{loc,tot}}(\Lambda^{\otimes n}(\hat{\rho}^{(n)}); \hat{H}^{(n)})}{n} ,$$

and the *separable-input local total-ergotropy capacitance*

$$C_{\text{sep,loc,tot}}(\Lambda; \mathfrak{e}) \tag{A16}$$
$$:= \limsup_{n\to\infty} \max_{\hat{\rho}_{\text{sep}}^{(n)}\in\mathfrak{S}_{E=n\mathfrak{e}}^{(n)}} \frac{\mathcal{E}_{\text{loc,tot}}(\Lambda^{\otimes n}(\hat{\rho}_{\text{sep}}^{(n)}); \hat{H}^{(n)})}{n} ,$$

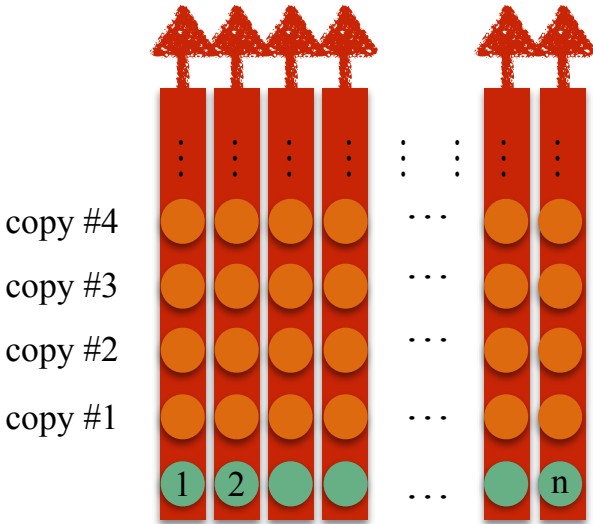

FIG. 5. Schematic representation of the type of operations implicitly allowed in the definition of $C_{\text{loc,tot}}(\Lambda; \mathfrak{e})$ and $C_{\text{sep,loc,tot}}(\Lambda; \mathfrak{e})$. In this scheme the green elements are the $n$ q-cells that compose the quantum battery (they can be initialized either in correlated quantum states or in separable ones). The orange elements represent identical copies of the quantum battery (ideally we have an infinite number of them). The agent is allowed to operate on the system by acting with unitaries (red vertical elements in the figure) that may couple a given q-cell with all its copies; however, couplings connecting a given q-cell and its copies with a different q-cell and its copies are not allowed.

Formally speaking, $C_{\text{sep,loc,tot}}(\Lambda; \mathfrak{e})$ allows for unitary transformations that are *local* with respect to the different q-cells but *global* on the collections of copies of each one of them – see Fig. 5.

To derive this result consider the special case in which $\Lambda$ is a replacement channel inducing the transformation

$$\Lambda(\hat{\rho}) = \text{Tr}[\hat{\rho}]\, \hat{\rho}_0 \,, \qquad \forall \hat{\rho} \,, \tag{A17}$$

being $\hat{\rho}_0$ a fixed q-cell state. By construction any input state of $n$ q-cells $\hat{\rho}^{(n)}$ (included the separable configurations) will be transformed in the density matrix $\hat{\rho}_0^{\otimes n}$. Hence for all $n$ and $\hat{\rho}^{(n)}$ we have

$$\frac{\mathcal{E}_{\text{loc}}(\Lambda^{\otimes n}(\hat{\rho}^{(n)}); \hat{H}^{(n)})}{n} = \frac{\mathcal{E}_{\text{loc}}(\hat{\rho}_0^{\otimes n}; \hat{H}^{(n)})}{n} = \mathcal{E}(\hat{\rho}_0; \hat{h}) \,, \tag{A18}$$

which follows from the additivity of the local ergotropy for independent (i.e. non interacting) tensor product states, and

$$\frac{\mathcal{E}_{\text{tot}}(\Lambda^{\otimes n}(\hat{\rho}^{(n)}); \hat{H}^{(n)})}{n} = \frac{\mathcal{E}_{\text{tot}}(\hat{\rho}_0^{\otimes n}; \hat{H}^{(n)})}{n} = \mathcal{E}_{\text{tot}}(\hat{\rho}_0; \hat{h}) \,, \tag{A19}$$

which instead follows from the additivity property of the total ergotropy for independent tensor product states.

Similarly we also have

$$\begin{aligned} \frac{\mathcal{E}_{\text{loc,tot}}(\Lambda^{\otimes n}(\hat{\rho}^{(n)}); \hat{H}^{(n)})}{n} &= \frac{\mathcal{E}_{\text{loc,tot}}(\hat{\rho}_0^{\otimes n}; \hat{H}^{(n)})}{n} \\ &= \frac{\mathcal{E}_{\text{tot}}(\hat{\rho}_0^{\otimes n}; \hat{H}^{(n)})}{n} \\ &= \mathcal{E}_{\text{tot}}(\hat{\rho}_0; \hat{h}) \,, \end{aligned} \tag{A20}$$

where in the second identity we used the fact that the locality constraint on different q-cells is irrelevant when computing the total-ergotropy of tensor product states.

Replacing Eq. (A18) into Eqs. (45) and (46) gives

$$C_{\text{loc}}(\Lambda; \mathfrak{e}) = C_{\text{sep,loc}}(\Lambda; \mathfrak{e}) = \mathcal{E}(\hat{\rho}_0; \hat{h}) \,, \tag{A21}$$

while replacing Eq. (A20) in Eqs. (A15) and (A16) gives

$$C_{\text{loc,tot}}(\Lambda; \mathfrak{e}) = C_{\text{sep,loc,tot}}(\Lambda; \mathfrak{e}) = \mathcal{E}_{\text{tot}}(\hat{\rho}_0; \hat{h}) \,, \tag{A22}$$

(notice that from Eq. (A19) follows that $\mathcal{E}_{\text{tot}}(\hat{\rho}_0; \hat{h})$ also corresponds to the value of $C_{\mathcal{E}}(\Lambda; \mathfrak{e})$ and $C_{\text{sep}}(\Lambda; \mathfrak{e})$). Choosing $\hat{\rho}_0$ so that $\mathcal{E}(\hat{\rho}_0; \hat{h}) < \mathcal{E}_{\text{tot}}(\hat{\rho}_0; \hat{h})$ (a condition that can be fulfilled if $d > 2$) we can consequently show that there is a finite gap between the total-ergotropy versions of the local and separable-input local capacitances. In particular, by identifying $\hat{\rho}_0$ with a passive but not completely passive state of the system, we have

$$\begin{aligned} C_{\text{loc}}(\Lambda; \mathfrak{e}) &= C_{\text{sep,loc}}(\Lambda; \mathfrak{e}) = 0 \,, \tag{A23} \\ C_{\text{loc,tot}}(\Lambda; \mathfrak{e}) &= C_{\text{sep,loc,tot}}(\Lambda; \mathfrak{e}) > 0 \,. \tag{A24} \end{aligned}$$

### 3. Equivalence between ergotropic MAWER and total ergotropy MAWER

Here we show that the MAWER defined in Eq. (49) can be computed by replacing the erogotropy with the total-ergotropy, i.e.

$$\mathcal{J}_{\mathcal{E}}(\Lambda) = \mathcal{J}_{\text{tot}}(\Lambda) := \limsup_{E \to \infty}\left( \sup_{n \geq 1} \frac{\mathcal{E}_{\text{tot}}^{(n)}(\Lambda; E)}{E} \right) \,. \tag{A25}$$

Since for each $n$ and $E$ we have $\mathcal{E}_{\text{tot}}^{(n)}(\Lambda; E) \leq \mathcal{E}^{(n)}(\Lambda; E)$, it is clear that $\mathcal{J}_{\mathcal{E}}(\Lambda) \leq \mathcal{J}_{\text{tot}}(\Lambda)$: thus to prove Eq. (A25) we only need to show that also the opposite is true. For this purpose consider $\hat{\rho}^{(n)} \in \mathfrak{S}_E^{(n)}$ and observe that

$$\begin{aligned} \frac{\mathcal{E}_{\text{tot}}(\Lambda^{\otimes n}(\hat{\rho}^{(n)}); \hat{H}^{(n)})}{E} &= \lim_{N \to \infty} \frac{\mathcal{E}(\Lambda^{\otimes nN}((\hat{\rho}^{(n)})^{\otimes N}); \hat{H}^{(nN)})}{NE} \\ &\leq \lim_{N \to \infty} \frac{\mathcal{E}^{(nN)}(\Lambda; NE)}{NE} \\ &\leq \lim_{N \to \infty} \sup_{M \geq 1} \frac{\mathcal{E}^{(M)}(\Lambda; NE)}{NE} \\ &\leq \limsup_{E' \to \infty} \sup_{M \geq 1} \frac{\mathcal{E}^{(M)}(\Lambda; E')}{E'} \\ &= \mathcal{J}_{\mathcal{E}}(\Lambda) \,. \tag{A26} \end{aligned}$$

Taking the max over all possible states $\hat{\rho}^{(n)} \in \mathfrak{S}_E^{(n)}$ we then obtain:

$$\frac{\mathcal{E}_{\text{tot}}^{(n)}(\Lambda; E)}{E} \leq \mathcal{J}_{\mathcal{E}}(\Lambda) , \qquad (A27)$$

which is valid for all $n$ and $E$. Then, taking first the sup over $n$ and successively the limsup over $E$, we finally obtain

$$\mathcal{J}_{\text{tot}}(\Lambda) \leq \mathcal{J}_{\mathcal{E}}(\Lambda) , \qquad (A28)$$

that gives the thesis. We remark that the derivation applies also in those cases where $\sup_{n\geq 1} \frac{\mathcal{E}^{(n)}(\Lambda;E)}{E}$ and $\sup_{n\geq 1} \frac{\mathcal{E}_{\text{tot}}^{(n)}(\Lambda;E)}{E}$ diverge.

## Appendix B: Computation of $\overline{\mathcal{E}}^{(1)}(\mathcal{D}_\lambda; E)$ and $\overline{\mathcal{E}}_{\text{tot}}^{(1)}(\mathcal{D}_\lambda; E)$ for depolarizing channels

Here we derive Eq. (81). For this purpose observe that, being $|\psi\rangle$ a pure state with input mean energy equal to $E$, the spectrum of $\hat{\rho}' := \mathcal{D}_\lambda(|\psi\rangle\langle\psi|)$ of Eq. (78) has two distinct eigenvalues:

$$\begin{aligned} \lambda_1 &:= \lambda + (1-\lambda)/d \quad \text{(non degenerate)}, \\ \lambda_2 &:= (1-\lambda)/d \quad \text{(with degeneracy } d-1\text{), (B1)} \end{aligned}$$

so that

$$\hat{\rho}' = \lambda_1 |\psi\rangle\langle\psi| + \lambda_2 \hat{\Pi}_\perp , \qquad (B2)$$

where $\hat{\Pi}_\perp := \hat{\mathbb{1}} - |\psi\rangle\langle\psi|$ is the projector on the $d-1$ subspace orthogonal to $|\psi\rangle$. Notice also that for $\lambda \geq 0$ we have $\lambda_1 \geq \lambda_2$. From Eq. (3) it hence follows that in this case the passive counterpart of $\hat{\rho}'$ is the density matrix

$$\begin{aligned} \hat{\rho}'_{\text{pass}}(\geq) &:= \lambda_1 |\epsilon_1\rangle\langle\epsilon_1| + \lambda_2 \sum_{\ell=2}^{d} |\epsilon_\ell\rangle\langle\epsilon_\ell| \\ &= (\lambda_1 - \lambda_2)|\epsilon_1\rangle\langle\epsilon_1| + \lambda_2 \hat{\mathbb{1}} , \quad (B3) \end{aligned}$$

which has mean energy $\mathfrak{E}(\hat{\rho}'_{\text{pass}}(\geq); \hat{H}) = \lambda_2 \text{Tr}[\hat{h}]$ (remember that the eigenvalues of $\hat{h}$ obey the ordering

$0 = \epsilon_1 \leq \epsilon_2 \leq ... \leq \epsilon_d = 1$). Accordingly in this case the output ergotropy of the state $|\psi\rangle$ writes

$$\begin{aligned} \mathcal{E}(\hat{\rho}'; \hat{h}) &= \mathfrak{E}(\hat{\rho}'; \hat{H}) - \lambda_2 \text{Tr}[\hat{h}] \qquad (B4) \\ &= \lambda E + \frac{(1-\lambda)}{d}\text{Tr}[\hat{h}] - \lambda_2 \text{Tr}[\hat{h}] = \lambda E , \end{aligned}$$

which, being independent from the specific choice of $|\psi\rangle$, leads to Eq. (81) for $\lambda \geq 0$. For $\lambda < 0$ the role of $\lambda_1$ and $\lambda_2$ gets inverted with $\lambda_2$ becoming the largest, i.e. $\lambda_2 \geq \lambda_1$. Accordingly Eq. (B3) must be replaced by

$$\begin{aligned} \hat{\rho}'_{\text{pass}}(<) &:= \lambda_2 \sum_{\ell=1}^{d-1} |\epsilon_\ell\rangle\langle\epsilon_\ell| + \lambda_1 |\epsilon_d\rangle\langle\epsilon_d| \qquad (B5) \\ &= (\lambda_1 - \lambda_2)|\epsilon_d\rangle\langle\epsilon_d| + \lambda_2 \hat{\mathbb{1}} = -\lambda|\epsilon_d\rangle\langle\epsilon_d| + \lambda_2 \hat{\mathbb{1}} , \end{aligned}$$

leading to $\mathfrak{E}(\hat{\rho}'_{\text{pass}}(<); \hat{H}) = -\lambda + \lambda_2 \text{Tr}[\hat{h}]$ and

$$\begin{aligned} \mathcal{E}(\hat{\rho}'; \hat{h}) &= \lambda E + \frac{(1-\lambda)}{d}\text{Tr}[\hat{h}] + \lambda - \lambda_2 \text{Tr}[\hat{h}] \\ &= \lambda(E-1) = |\lambda|(1-E) , \qquad (B6) \end{aligned}$$

which proves Eq. (81) for $\lambda < 0$.

The value of the single-site maximum output total ergotropy term $\mathcal{E}_{\text{tot}}^{(1)}(\mathcal{D}_\lambda; E)$ proceeds in a similar fashion. The starting point is the observation that for all pure input states $|\psi\rangle$ we have that the corresponding output entropy doesn't depend on the input energy $E$, i.e.

$$S(\hat{\rho}') = S_d(\lambda) := -\lambda_1 \log_2 \lambda_1 - \lambda_2(d-1)\log_2 \lambda_2 , \quad (B7)$$

with $\lambda_{1,2}$ defined as in Eq. (B1). Given hence

$$Z_\beta(\hat{h}) := \text{Tr}[e^{-\beta\hat{h}}] = \sum_{\ell=1}^{d} e^{-\beta\epsilon_\ell} , \qquad (B8)$$

the partition function of a single q-cell, and

$$\mathfrak{E}_{\text{GIBBS}}^{(\beta)}(\hat{h}) := -\frac{d}{d\beta}\ln Z_\beta(\hat{h}) , \qquad (B9)$$

$$S_\beta := -\beta\frac{d}{d\beta}\ln Z_\beta(\hat{H}) + \ln Z_\beta(\hat{H}) , \qquad (B10)$$

the associated Gibbs state mean energy and entropy, we can write

$$\begin{aligned} \mathcal{E}_{\text{tot}}(\hat{\rho}'; \hat{h}) &= \mathfrak{E}(\hat{\rho}'; \hat{h}) - \mathfrak{E}_{\text{GIBBS}}^{(\beta_\star)}(\hat{h}) \qquad (B11) \\ &= \lambda E + \frac{(1-\lambda)}{d}\text{Tr}[\hat{h}] - \mathfrak{E}_{\text{GIBBS}}^{(\beta_\star)}(\hat{h}) , \end{aligned}$$

with $\beta_\star$ a function of $\lambda$ and $d$, obtained by solving the equation

$$S_{\beta_\star} = S_d(\lambda) . \qquad (B12)$$

Equation (83) finally follows by setting

$$D_{\text{tot}}(\lambda; \hat{h}) := \frac{(1-\lambda)}{d}\text{Tr}[\hat{h}] - \mathfrak{E}_{\text{GIBBS}}^{(\beta_\star)}(\hat{h}) . \qquad (B13)$$

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
