# Peer review of "Quantum Work Capacitances: ultimate limits for energy extraction on noisy quantum batteries"

_SciPost Physics_

## Round 1 · Referee Report · Anonymous (Referee 1) · 2023-5-5

Strengths
Weaknesses
Report
Warnings issued while processing user-supplied markup:
- Inconsistency: plain/Markdown and reStructuredText syntaxes are mixed. Markdown will be used.
Add "#coerce:reST" or "#coerce:plain" as the first line of your text to force reStructuredText or no markup.
You may also contact the helpdesk if the formatting is incorrect and you are unable to edit your text.
Report on Tirone et al's manuscript "Quantum Work Capacitances"
This could be an interesting work about the loss of work capacity in quantum batteries due to decoherence and relaxation processes, but I see many signs that it was written too fast. It has missing citations, inconsistent notation, strange choices for acronyms, and some sections are not proofread at all. However these are easily fixed, and they are not my main criticism.
My MAIN CRITICISM is that this manuscript uses formal mathematics to define quantities and derive results, but neither the quantities nor the results are discussed in term of their physics. The manuscript provides no physical motivation for the definition that it takes for "Quantum Work Capacitances" (which requires perfect individual control of an infinite number of quantum systems). It also provides no discussion of the physical consequences of its results. At the same time, the quantities and results are not compared with what is already known in the literature. This makes the manuscript unsuitable for a physics journal in its current form.
The reasons for my main criticism are the following:
(I) The title is about "quantum work capacitances" and the abstract promises that this work will introduce them. However, the definition of the work capacitance is only presented in terms of formal mathematics in section IIIA, with no physical explanation. The introduction gives no hint of what they are, or if similar quantities have been studied before. It took me some time to realise that it can be explained in simple words as follows; it is defined as the maximum capacity of a quantum battery to do work when it stores an energy E in an infinite number of systems, so each system only needs to store an infinitesimal amount of energy.
(II) When I googled for keywords like "quantum battery dissipation", I found a paper on the subject that was not cited by this manuscript; "Dissipative dynamics of an open quantum battery" M Carrega1, A Crescente, D Ferraro and M Sassetti, New Journal of Physics, Volume 22, August 2020
(III) The manuscript needs to place itself in the context of earlier works. It currently simply says that refs [5,13-21] ansd [22,23] worked on the same subject. The manuscript should summarize what is already known about loss of work capacity in quantum batteries due to decoherence and relaxation processes. What is the same or different in this manuscript compared to all these earlier works (and also the work of Caregal cited above)? For example, what measures did those earlier works use, and are the measures proposed in this work different or better?
(IV) The manuscript provides various mathematical proofs for the "quantum work capacitances", but does not explain what these proofs mean in terms of the physics of real systems. They have simple examples in term of two level systems, but even then theystay in the language of formal mathematics, and do not say in simple words what they have proven about the physics of such two-level systems. Here are just two examples:
(a) Eq. (60) seems to implies a clear physical consequence, that the battery provide the same maximum amount of work irrespective of whether there is entanglement between its quantum systems, and irrespective of whether one limits the extraction to local unitaries or not. BUT this is NOT explicitly stated.
(b) It looks from Eqs. (57,60,63) that we can only extract epsilon of work by unitary rotations, but you can extract MORE by coupling the battery to a bath, because then we can extract espsilon + Log Z/beta where Log Z is positive (as seen from the definition of Z below Eq (6)). BUT this is also NOT explicitly stated. Actually, this result SURPRISES ME GREATLY, I would have expected the opposite! Is it new, or is it already known? How can it be explained?
I am sure every result in this work has similar physical consequences. It is the authors job to give them, not leave the readers to struggling to figure them out for themselves.
(V) The manuscript defines quantities that they call "Quantum Work Capacitances" (the title of the paper) which are only defined in the limit of taking the number of systems to infinity. This is not the usual macroscopic limit, because one still need perfect control of each individual system (to perform an arbitrary unitary rotation on each system individually). The authors give no physical motivation for WHY this limit is interesting.
(a) WHY define the "work capacitance" as the infinite n limit? (b) Why not define it for n systems?
As a physicist, I know we will never be able to control an infinite number of systems perfectly, but I know that experiments exist with good individual control of tens of qubits (Google's Sycamore quantum computer, etc). So what can one learn about the physics with n=20 or n=30 from the infinite n "quantum work capacitances" presented here.
(VI) Finally the manuscript is hard to read, with poorly chosen or poorly defined notation, and strange choices for acronyms. Also some sections contain multiple typos. These are specific points that are probably easy to fix, and I list them below.
=========================
SPECIFIC POINTS:
What is the difference between N in Eq. (5) and n in Eq (25), why not use n throughout? Why is it "lim" in Eq. (5) and "lim sup" in Eq (25)?
I cannot understand the logic in the use of ${\cal E}$ and ${\cal W}$ in the manuscript. It seems to vary from paragraph to paragraph. Please clarify this notation. If ${\cal E}$ and ${\cal W}$ are different, please be clearer about how they differ! For example: (a) Why are all work extraction functionals called ${\cal E}$ except the free-energy one which is called ${\cal W}$? Why not call it ${\cal E}_\beta$ to be consistent? (b) The first sentence of section IIB should NOT say ${\cal W}$ was defined in the previous section, because it is not the same as defined there; it is not the ${\cal W}$ in Eq. (10)! Instead it needs to say "Here ${\cal W}$ stands for one of our four work extraction functionals (i.e. ergotropy, total ergotropy, non-equilibrium free energy and local ergotropy) defined in the previous section" (like a sentence later in the manuscript). (c) In Eq. (35) a new quantity ${\cal E}^{(n)}$ appears without definition. I think it is what is called ${\cal W}^{(n)}$ elsewhere in the work, but I am not sure!
Why does Eq. (10) have the second term "Log Z"? What is its physical meaning. In normal thermodynamics, the capacity to do work is the free energy F, indeed "free energy" is DEFINED as the capacity of a system to do work when coupled to a thermal bath. Thus the Log Z term here requires explanation.
I was initially very confused by the sentence "This is due to the possibility of super-additive effects" following the unnumbered equation below Eq (28). We all learn that systems that are non-interacting (as in the Hamiltonian below Eq. (21)), have additive free-energies. I think this is not true here, because the systems can start in an entangled state. If this is so, it would help the reader to write "This is due to the possibility of super-additive effects for quantum batteries made of quantum systems that are entangled with each other" (or similar).
Below Eq. (45) it is says "where $\hat{W}$ is the element ..." but $\hat{W}$ does not appears in Eq. (45), and the words "second passage" do not seem to refer to anything here (unless it means "second line").
At the beginning of section VA it says the "the QBs are two-level systems", it would be clearer as "the QB is made of two-level systems". Since the second paragraph is about a QB containing many two-level systems.
It is hard to read a paper which uses little known acronyms. I have to flick through the pages to find the definition of each one. As there is no limit to space in this paper, there is no reason to use any of the following acronyms. WE = Work extraction QB = quantum battery ER = ?? Note that "ER" was used throughout manuscript in "ER process", "ER form" and "ER efficiency", but I lost patience when looking for the definition of this, I guess it is somewhere in the manuscript, but I do not know where! Replacing these three acronyms with the full words will make the text much easier to understand.
Also the manuscript uses the acronym "LCPT map". I think this is the same as "CPTP map" which is a standard acronym that all expert readers ALREADY know (e.g. a google search for "CPTP" has hits, but "LCPT" has none). I think the word "linear" is omitted because quantum mechanics is linear and so linearity can be assumed. If the LCTPT map is different from the usual CPTP map because of the linearity, this should be explained, but then why drop the P for "Preserving", why not use "LCPTP"?
There are a cluster of typos of page 7 second column: "numeber" should be "number" "elements let's" should be "elements, let us" "For arbitrary n an E" should be for "For arbitrary n and E" The authors should check for other clusters of typos.
Requested changes
See my report

---

## Round 1 · Referee Report · Anonymous (Referee 2) · 2023-7-9

Strengths
Weaknesses
Report
In the manuscript, authors propose a set of measures to probe the maximum efficiency in extracting work out of a quantum system. Authors give a detailed mathematical framework, introduce the proposed quantities, investigate the asymptotic limits and some useful properties of them. Finally, they applied the measures to a qubit system with two different noise models. The manuscript contains useful results to quantify how a selected state of a quantum battery is resilient to the detrimental action of a given noise model as well as to identify the optimal initial state for the energy recovery task.
Although the work seems interesting, the manuscript is very confusing to read and needs to be revised heavily before consideration for publication. With a better and more accessible conveying of the proposed ideas, it could potentially serve as a good work. I listed my main comments and criticisms below:
1) In Eq. (10) it seems like free energy of a specific state (first term) and the global free energy (second term) is added together. Why? Doesn’t the contribution of the specific state already included in the global free energy? The same goes for Eq. (6). Can authors explain the logic behind those formulas?
2) I find the manuscript to be too technical. Many concepts are introduced but not justified properly. Why are the proposed measures (capacitances and MAWER) defined in those ways? Are the definitions general/universal or system/condition specific? What are the physical justifications? In addition to clarify these points, as a general comment, the authors should consider providing a more accessible introduction to these concepts to make the paper more accessible to a broader audience. Although authors have already used the Appendix to reduce the mathematical load, I still think they could use even more. Making a concise paper focusing on the physics of the problem and putting detailed derivations, even the properties to Appendix could increase the focus and clarity much more than its current form.
3) Authors provide new measures and approaches but there is no proper comparison given with the existing methods. They mention other works on this as references, however a more detailed comparison should be given. How are they related to the existing methods? Why we need new ones? How the proposed approaches are different? How are they better or worse in various ways? Pros&cons etc. Too many new concepts are introduced in the paper. Each of them needs to be justified properly, as this is the main novelty of the work.
4) I find the term “work capacitance” to be a bad choice. Capacitance has a well-established meaning in physics, and I do not think what is proposed in the manuscript relates to that. This discrepancy may lead to confusion and misinterpretation among readers. “Work capacity” might be a better term for the proposed measures.
5) I have a similar comment about the title. It is generally suggested to avoid using newly introduced terms in the title as it does not connote anything in people’s minds. The title in its current form says not much about the content of the manuscript. People will hesitate to read something they don’t understand right away. I suggest authors to reconsider the title. Defining new figures of merit could be mentioned in the title.
6) A qubit system is chosen for the application. However, the limitations of the measures are not properly discussed. Can they be applied to other quantum systems? Are the results general? If yes, it should be explicitly shown and emphasized, which would increase the impact of the work.
Apart from these, there are other simpler issues about the proofreading:
1) After Eq. (10) in the definition of non-equilibrium free energy, S should be divided by T not β. I did not check all the equations one by one, I would double check whether all equations are correct.
2) I find some of the mathcal notation and the choice of characters to be weird. There are unnecessary usage of acronyms. “WE” should be written in full form and “ER” is never defined. I do not know what it is.
3) There are many typos: e.g. eingenvalues.
4) Authors should use genderless pronouns rather than male pronouns. E.g. “…he can perform...”
Requested changes
See the report.

---

## Round 2 · Referee Report · Anonymous · 2024-5-27

Strengths

The results in this work are very interesting, and the manuscript now nicely explains those results.

Weaknesses

None

Report

The authors have fully addressed my comments, so I am happy to recommend this work for publication.
I think it is a very nice paper.

Requested changes

None

Recommendation

Publish (surpasses expectations and criteria for this Journal; among top 10%)

---

## Round 2 · List of Changes

We thank the referees and the editor.

As one of the referees requested, we have modified the paper in the following way:

- We have added a small summary of results at the beginning of section 5.
- We have now written in page 5 all the explicit definitions for all the quantities the we used in the article, and we have clarified their meaning and the relations among them.
- We have corrected the typos.
- We have reformulated part of the discussion on the results on the depolarizing channel to answer a question raised by the referee.

---

## Editorial Decision

accepted_in_target_journal